# DNA polymerase ε relies on a unique domain for efficient replisome assembly and strand synthesis

Xiangzhou Meng [1], Lei Wei[1,3], Sujan Devbhandari[1], Tuo Zhang [2], Jenny Xiang[2], Dirk Remus [1] & Xiaolan Zhao [1]✉

DNA polymerase epsilon (Pol ε) is required for genome duplication and tumor suppression. It supports both replisome assembly and leading strand synthesis; however, the underlying mechanisms remain to be elucidated. Here we report that a conserved domain within the Pol ε catalytic core influences both of these replication steps in budding yeast. Modeling cancer-associated mutations in this domain reveals its unexpected effect on incorporating Pol ε into the four-member pre-loading complex during replisome assembly. In addition, genetic and biochemical data suggest that the examined domain supports Pol ε catalytic activity and symmetric movement of replication forks. Contrary to previously characterized Pol ε cancer variants, the examined mutants cause genome hyper-rearrangement rather than hyper-mutation. Our work thus suggests a role of the Pol ε catalytic core in replisome formation, a reliance of Pol ε strand synthesis on a unique domain, and a potential tumor-suppressive effect of Pol ε in curbing genome re-arrangements.

[1] Molecular Biology Program, Memorial Sloan Kettering Cancer Center, New York, NY 10065, USA. [2] Genomics Resources Core Facility, Weill Cornell Medical College, New York, NY 10065, USA. [3] Present address: Department of Molecular Biology, Princeton University, Princeton, NJ 08544, USA. ✉email: zhaox1@mskcc.org

Eukaryotic genome duplication requires three conserved replicative polymerases. Among them, DNA polymerase epsilon (Pol ε) carries out the bulk of leading strand synthesis[1–5]. Its large subunit, POLE in mammals and Pol2 in budding yeast, supplies the catalytic activity and its three other subunits have structural roles[6–10]. Pol2 and its orthologs synthesize DNA with high fidelity due to their nucleotide selectivity and exonuclease (EXO) domain-mediated proofreading[11]. Pol2 family enzymes are almost twice the size of the catalytic subunits of other replicative polymerases (Pol α and δ), suggesting additional roles beyond DNA polymerization. Data on mammalian POLE is limited, but yeast studies have suggested that Pol2 is unique among replicative polymerase for its involvement in replisome assembly during S phase and activation of the replication checkpoint in genotoxic stress[12–14]. Elucidating how Pol2 executes these important and distinct roles is critical for a mechanistic understanding of genome inheritance.

The current model postulates that the Pol2 N-terminal half (Pol2-NT) harboring polymerase and EXO domains carries out strand synthesis, whereas its C-terminal structure half (Pol2-CT) not found in other DNA polymerases supports replisome assembly and checkpoint activation (Supplementary Fig. 1a)[15–17]. Intriguingly, Pol2-NT contains several highly conserved domains that are absent in other DNA polymerases, only one of which (P-domain) has been examined so far[18]. Moreover, the Pol2-NT, unlike Pol2-CT, adopts dynamic conformations, raising the question of whether Pol2-family-specific domains within this region contribute to distinct tasks at different stages of replication[17,19,20]. Given the fundamental importance of Pol2 and its orthologs in genome duplication, addressing this question is critical for deriving eukaryotic replication models.

Here, we examine an uncharacterized Pol2 catalytic core region uniquely possessed by Pol2 orthologs. To address how perturbation of this region affects DNA replication, we modeled recurrent cancer-associated POLE mutations found there based on the following rationale. It is well known that EXO mutations in POLE can drive tumorigenesis of hyper-mutated cancers[21–24]. However, recent analyses found frequent incidence of non-EXO POLE mutations in non-hyper-mutated cancers and a potential POLE contribution to the etiology of these cancers, arguing for the importance of non-EXO POLE variants in tumorigenesis[23,24]. Given that non-EXO POLE variants remain largely untested, we reasoned that modeling them in yeast could not only advance our understanding of wild-type Pol ε functions, but also inform us on genome disruptive potentials of non-EXO Pol ε mutations.

Applying this strategy in our study yields several insights. Our molecular, genetic, and biochemical data suggest a structural role of the Pol2 catalytic core in replisome assembly. Moreover, we find that the examined region specifically possessed by Pol2 family proteins has a previously unrecognized effect on DNA strand synthesis. Interestingly, we uncover Pol2 variants that induce large genomic changes without affecting mutation rates. This work sheds light on the mechanisms of replisome assembly and replicative DNA synthesis and expands our views on tumor-suppressive potentials of POLE.

## Results

**A unique domain of Pol2-family proteins is essential.** We examined a region of sixty-eight amino acids that is positioned at the periphery of the Pol2 catalytic core, away from its DNA binding and active sites (Supplementary Fig. 1a)[18]. This region shows 71% sequence homology between yeast Pol2 and human POLE but is not found in other types of DNA polymerases (Fig. 1a and Supplementary Fig. 1a). We refer to this region as POPS (POl2 family-specific catalytic core Peripheral Subdomain)

hereafter. To address the functions of POPS, we introduced mutations of conserved residues by modeling POLE changes found in cancer cells (Supplementary Fig. 1a and Table 1)[22,23]. Simultaneous substitution of five residues (R567C, K593C, S595P, E611K, L621F) caused lethality in plasmid shuffle experiments, suggesting that POPS is essential (Supplementary Fig. 1b). When only three POPS residues were mutated (R567C, E611K, L621F), cells were viable at lower temperatures, but not at 37 °C (Supplementary Fig. 1b). We confirmed this using an integrated allele of pol2-R567C, E611K, L621F (pol2-REL) that replaced the wild-type POL2 (Fig. 1b). We found that pol2-REL, which did not affect Pol2 protein levels, slowed S phase entry and progression (Fig. 1c and Supplementary Fig. 1c). These data suggest that POPS is critical for DNA replication.

**POPS is required for DNA synthesis throughout the genome.** We subjected pol2-REL cells to a wide-range of assays to determine the molecular consequences of POPS perturbation. Each individual mutation in this allele was also examined, as described in later sections, to determine the effects of cancer-associated mutations. We first generated DNA replication profiles of synchronized S phase cells using deep sequencing and copy number measurements[25,26]. Even at the permissive temperature for pol2-REL (24 °C), DNA synthesis was reduced throughout the genome in the mutant compared to wild-type cells (Supplementary Fig. 1d). We note that replication origin usage and timing were not altered in pol2-REL cells (Supplementary Fig. 1d). To gain a global view of early verse late origin behavior, we performed meta-analyses of >200 replication origins. Reduced DNA synthesis was found at regions containing both early and late origins in pol2-REL cells, with the latter exhibiting greater defects (Fig. 1d). These findings suggest that POPS is important for DNA synthesis throughout the genome.

**POPS affects replication initiation and fork movement.** To further delineate the effects of POPS on DNA replication, we performed two-dimensional agarose gel electrophoresis (2D gel, Supplementary Fig. 2a). We first examined the late origin ARS1212 located at the mid-point of the restriction fragment tested, such that bi-directional replication initiated there is expected to produce bubble-shaped replication intermediates (RIs) (Supplementary Fig. 2b, top). When wild-type cells were released from G1 arrest into S phase at 24 °C, bubble RIs were detected at 30 min, peaked at 40–50 min, and disappeared at 60 min (Fig. 1e). However, the temporal pattern of bubble RI formation in pol2-REL cells was reproducibly delayed (Fig. 1e). Such a defect is suggestive of a replication initiation deficiency.

Strikingly, we observed that pol2-REL, but not wild-type cells, exhibited abundant large Y-shaped RI signals coinciding with replication initiation at ARS1212 (Fig. 1e). For DNA fragments containing centrally localized origins, large Y-shaped RIs can be generated by the asymmetric movement of sister replication forks (Supplementary Fig. 2b, bottom). We note that small Y-shaped RIs produced by passive replication from neighboring origins were at comparable levels in wild-type and pol2-REL cells, consistent with normal origin usage and timing in the mutant (Fig. 1e and Supplementary Fig. 1d). Moreover, we observed that bubble and large Y-shaped RIs persisted even at 120 min after G1 release in pol2-REL cells, whereas wild-type replication was largely completed by 60 min (Fig. 1e and Supplementary Fig. 2c), suggesting that pol2-REL cells suffer fork movement defects.

Next, we assessed whether defective replication features seen for late origins were also manifested in early origins. To this end, DNA samples examined above were assayed for early origin

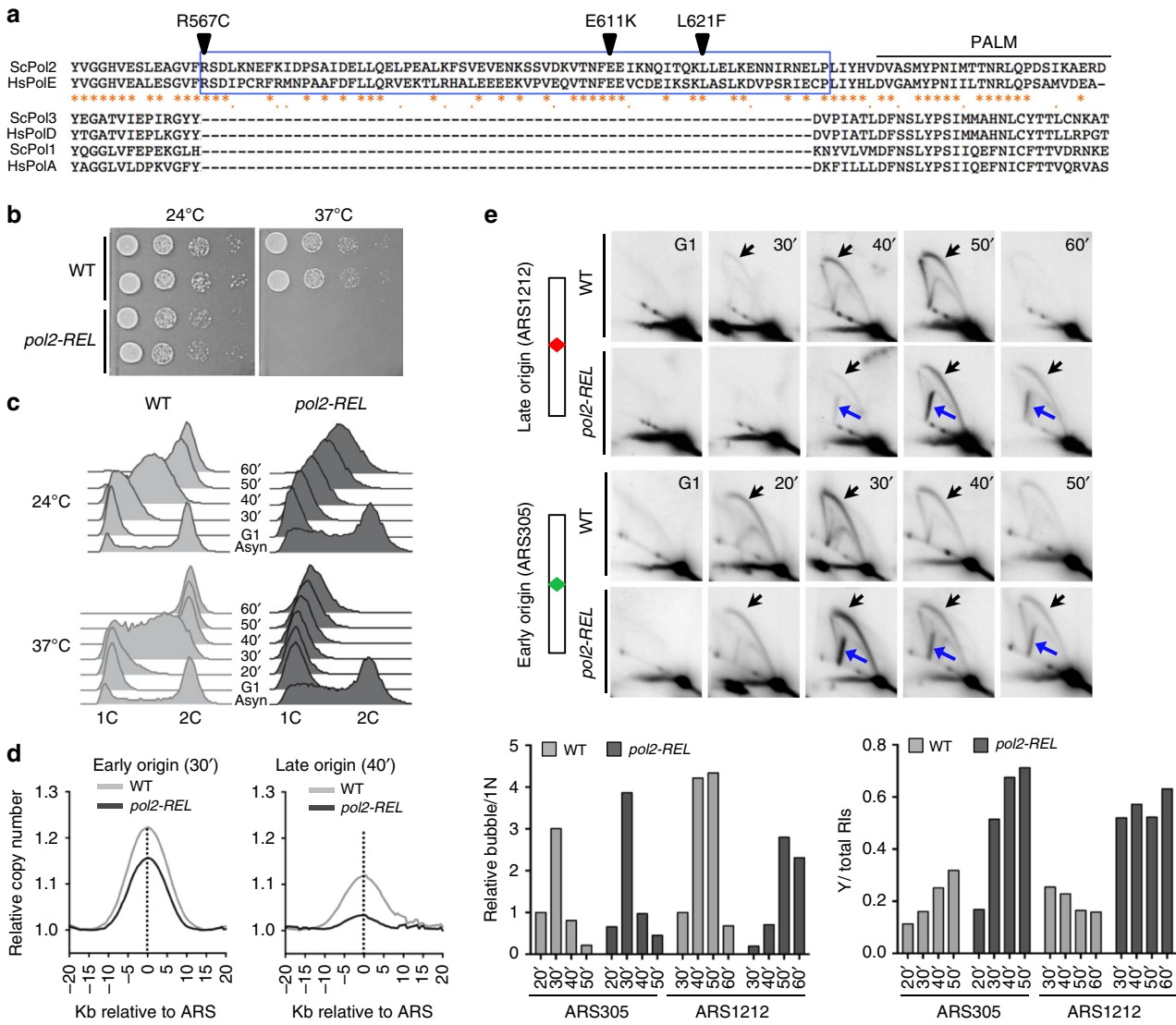

**Fig. 1 Mutating a Pol2 catalytic core domain impairs replication initiation and progression. a** Sequence alignment of POPS and adjacent regions in replicative polymerases. Sequences of POPS are boxed blue for the budding yeast Pol2 (ScPol2) and human POLE (hsPOLE) and absent in the catalytic subunits of DNA polymerase α (ScPol1, HsPolA) and δ (ScPol3, HsPolD). Regions adjacent to POPS, including the PALM domain, share homology among all replicative polymerases. Asterisks and dots label conserved and similar residues, respectively. Triangles highlight mutations in *pol2-REL*. **b** *pol2-REL* cells exhibit growth impairment. Tenfold serial dilutions of mutant and wild-type (WT) cells in biological replicates were spotted on plates and grown at the indicated temperatures. **c** Flow cytometry profiles suggest replication defects in *pol2-REL* cells. G1 synchrony was achieved by alpha-factor treatment of asynchronous culture (Asyn) at 24 °C. Flow cytometry monitored cellular DNA content upon release from G1 arrest into cycling at either 24 °C or 37 °C. **d** A meta-analysis of relative DNA copy numbers based on genome-sequencing results. Wild-type and *pol2-REL* cells were examined at 30′ and 40′ post G1-release at 24 °C as in panel c. Twenty kilo-bases from either side of early origins (n = 97, left) or late origins (n = 174, right) were averaged for copy numbers and plotted against relative positions of origins. The dotted line indicates the mid-point of origins. **e** Two-dimensional (2D) gel analyses reveal defective replication initiation and progression in *pol2-REL* cells. Samples from panel d were tested and 2D gel results for ~6 kb region containing the late origin ARS1212 or the early origin ARS305 are shown. The mid-point localization of the origins (diamonds) in the restriction fragments is shown on the side. Blue arrows signify the Y-shaped replication intermediates (RIs) in *pol2-REL* cells and black arrows label bubble-shaped RIs. Quantification of bubble- and Y-shaped RIs in WT and *pol2-REL* cells are shown at the bottom. For the former, the level of bubble-shaped RIs in WT at 20′ (for ARS305) or 30′ (for ARS1212) was set at 1. Delayed appearance of bubbled RIs and increased levels of Y-shaped RIs in *pol2-REL* cells were reproducibly detected in multiple trials using additional spore isolates. Signals of the bubble structures were normalized to 1N DNA to derive the percentage of bubble structures of all time points. Source data are provided as a Source data file.

behaviors by stripping the membranes and re-probing for two early origins, ARS305 and ARS315. A replication initiation delay was seen only at ARS315 (Fig. 1e and Supplementary Fig. 2d), consistent with replication profiling data showing overall mild defects at early origins (Fig. 1d). Importantly, large Y-shaped signals were abundant at both early origins in *pol2-REL* but not wild-type cells (Fig. 1e and Supplementary Fig. 2d), suggesting

that asymmetric fork movement occurs at early origins regardless of initiation delay.

In summary, replication profiling and 2D gel analyses suggest that *pol2-REL* leads to delayed replication initiation preferentially at late origins and asymmetric replication fork (or fork) movement at early and late origins. These unique features highlight the utility of *pol2-REL* in understanding replication initiation and elongation.

**Table 1 Cancer-associated POLE mutations at POPS examined in this study.**

| POLE mutation (corr. Pol2 mutation) | Number of patients[a] | Cancer type[b] |
|---|---|---|
| R553 → C/H/S/L (*pol2-R567C*) | Five/Six | Cutaneous Melanoma, Lung Adenocarcinoma |
| E597 → K (*pol2-E611K*) | Two/Two | Lung carcinoma Head and Neck Cancer |
| L607 → F (*pol2-L621F*) | One/One | Cutaneous Melanoma |
| R579 → C/H (*pol2-K593C*) | Five/None | Urothelial Carcinoma, Myeloid Neoplasm Colon Adenocarcinoma, Pancreatic Adenocarcinoma |
| A581 → T/V (*pol2-S595P*) | Four/Four | Stomach Adenocarcinoma, Lung Adenocarcinoma Thyroid Cancer, Basal Cell Carcinoma |

The listed mutations are the only POLE mutations found in patients[23] and none is associated with hyper-mutations[22].
[a]The first numbers are based on cBio Portal data and the second ones are based on the report from Campbell et al.[22] Only conserved residues are shown.
[b]Based on cBio Portal data.

**POPS influences the levels of the CMG replicative helicase.**
Next, we addressed how POPS contributed to replication initiation. A hallmark of replication initiation is the de novo assembly of the replisome, including the assembly of Cdc45, Mcm2-7, and the GINS complex (Psf1-3 and Sld5) into the CMG replicative helicase. Using co-immunoprecipitation (co-IP), we confirmed simultaneous Cdc45 and Psf1 association with MCM in S, but not in G1 phase samples in wild-type cells (Fig. 2a and Supplementary Fig. 2e)[27]. Significantly, this association was reduced in *pol2-REL* cells at both 37 °C and 24 °C, with the former condition causing stronger defects (Fig. 2a and Supplementary Fig. 2e). Compromised CMG levels are consistent with the replication initiation defects observed in *pol2-REL* cells as described above and point to a potential role for POPS in CMG assembly.

**POPS mediates efficient Pol ε incorporation into the pre-LC.**
Pol ε has been implicated in CMG assembly via the formation of a four-member pre-loading complex (pre-LC), which also includes the Dpb11 and Sld2 scaffold proteins and GINS[28]. According to current models, the pre-LC delivers Pol ε and GINS to origin-bound MCM and Cdc45, leading to the assembly of CMG and the association of Pol ε with it, while Dpb11 and Sld2 are recycled to form additional pre-LCs[13,29]. It is unclear if these proteins can also support CMG formation by other means beyond the pre-LC.

We first examined whether *pol2-REL* affected pre-LC formation. We confirmed that in wild-type cells, Sld2 co-immunoprecipitated the pre-LC members Pol2, Dpb11, and Psf1 in S, but not G1, phase (Fig. 2b)[28]. Strikingly, Sld2 pulled down little Pol2-REL protein during S phase, even though Dpb11 and Psf1 were efficiently pulled down (Fig. 2b). Defective interaction between Pol2-REL and Dpb11 was also seen in Pol2 co-IP experiments (Fig. 2c). In contrast, the Pol ε subunit Dpb2 associated normally with Pol2-REL (Fig. 2c). Moreover, subunit stoichiometry was unaffected in purified Pol ε[REL] (Supplementary Fig. 2f). Thus, Pol2-REL is able to form Pol ε but is defective in interacting with other pre-LC components. This phenotype indicates that POPS affects pre-LC formation. The correlation of reduced pre-LC and CMG levels in *pol2-REL* cells supports the model that the pre-LC is critical for CMG assembly, but does not exclude other means by which Pol ε may affect CMG assembly.

**Dpb11 overexpression rescues several *pol2-REL* defects.** Given the impaired pre-LC formation seen in *pol2-REL* cells, we asked if increasing dosages of specific pre-LC members could rescue *pol2-REL* defects. Strikingly, overexpression of Dpb11, but not Sld2, suppressed *pol2-REL* lethality at 37 °C (Fig. 3a and Supplementary Fig. 3a). This rescue is associated with improved replication initiation and genome duplication, as assessed by 2D gel and

FACS analyses, respectively (Fig. 3b and Supplementary Fig. 3b). The improvement also correlated with a rescue of CMG levels (Fig. 3c). Significantly, increased Dpb11 dosages restored its association with Pol2-REL to a level similar to that seen with wild-type Pol2 (Fig. 3d). The simplest interpretation of these findings is that restoration of Pol2-REL interaction with Dpb11 improves CMG formation, and consequently replication initiation and growth.

We further tested how Dpb11 overexpression affects pre-LC assembly. Intriguingly, increased Dpb11 levels boosted Dpb11 and Sld2 association, but was insufficient for enhancing interactions between Pol2-REL and Sld2 or between Dpb11 and GINS (Fig. 3d and Supplementary Fig. 3c). That enhanced Pol2-REL and Dpb11 association without restoring pre-LC levels is sufficient to mitigate *pol2-REL* defects argues for pre-LC independent means in supporting CMG assembly, at least in certain conditions.

**Low pre-LC levels do not cause asymmetric fork structures.** After elucidating POPS's effect on replication initiation, we queried its influence on replication fork behaviors. We considered three possible explanations for the accumulation of asymmetric replication structures in *pol2-REL* cells and tested each using multiple approaches. First, low pre-LC levels in *pol2-REL* cells may result in unidirectional origin firing due to asymmetric assembly of a single replisome at the replication origin. To test this possibility, we asked whether lowering pre-LC levels *per se* could lead to asymmetric replication structures, as seen in *pol2-REL* cells. As Dpb11 is involved in pre-LC formation but does not travel with replication forks, its acute depletion in G1 and S phase is expected to primarily affect pre-LC levels[29]. Thus, we transcriptionally downregulated Dpb11 in G1 and S phase cells (Fig. 4a). As expected, Dpb11 loss reduced replication initiation structures (bubbles) at ARS305 and ARS1212, and slowed S phase; however, asymmetric replication intermediates were not observed (Fig. 4b, c and Supplementary Fig. 4a). This finding suggests that lowering pre-LC levels *per se* does not induce asymmetric origin firing, indicating that *pol2-REL* likely affects replication fork progression.

**Pol ε[REL] impairs DNA synthesis and replication initiation.** We then tested the second possibility that Pol ε[REL] may impair DNA strand synthesis. To this end, we employed the reconstituted origin-dependent yeast DNA replication system, performing both Pol ε titration experiments and time-course analyses[30]. When Pol ε was added at concentrations up to 30 nM, fork progression rates were largely unaffected by Pol ε[REL], as evident from the similar length distribution of leading strand products obtained with Pol ε[wt] or Pol ε[REL] (Fig. 4d and Supplementary Fig. 4b). However, at a higher Pol ε concentration (60 nM), moderate but reproducible

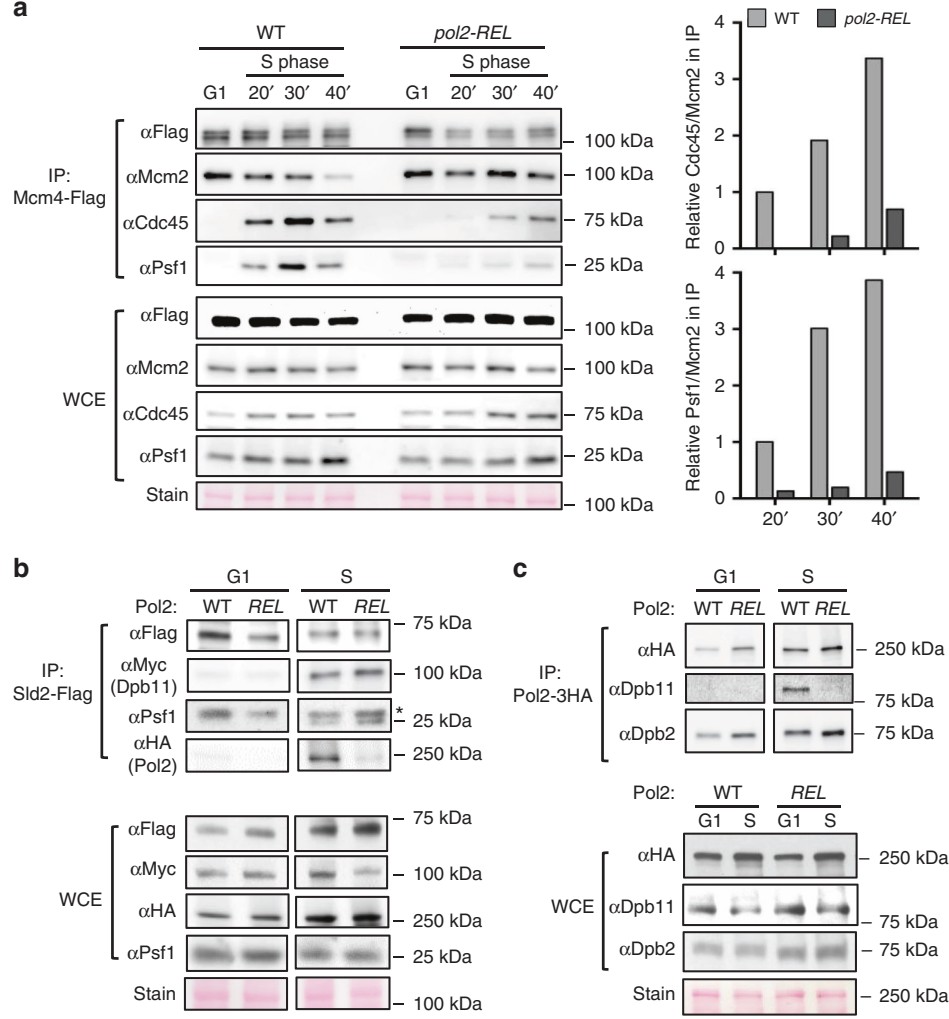

**Fig. 2 pol2-REL disrupts Pol2 incorporation into the pre-LC and CMG formation. a** CMG levels in wild-type and pol2-REL cells. Cells were synchronized in G1 at 24 °C and released into S phase at 37 °C. Flag-tagged Mcm4 was immunoprecipitated (IP), and the co-purification of Mcm2, Cdc45, and Psf1 was examined. Representative western blots are presented and quantification (right) of Cdc45 and Psf1 levels relative to those of Mcm2 in the IP fractions show a reduction in pol2-REL cells compared with wild-type cells. **b** Examination of pre-LC levels. Flag-tagged Sld2 was IP from protein extracts in G1 or S phase cells at 24 °C, and co-purified Pol2, Dpb11, and Psf1 were examined. The asterisk indicates the light chain of the antibody used in the IP. WCE: whole-cell extract. Stain serves as a loading control. **c** Examination of Pol2 association with Dpb11 and Dpb2. Experiments were done as in panel b, except that HA-tagged Pol2 was IP and Dpb11 and Dpb2 were examined. For all panels, similar results were obtained using at least two independent strains per genotype. Source data are provided as a Source data file.

reductions of both leading and lagging strand lengths were seen in reactions containing Pol ε^REL compared to Pol ε^wt (Fig. 4d and Supplementary Fig. 4b), indicating a fork progression defect in the presence Pol ε^REL. The concentration-dependent effect seen here could be due to Pol δ compensating for leading strand synthesis defects only when Pol ε^REL levels are low[30]. To further test if Pol ε^REL is defective for DNA synthesis, we assessed Pol ε^REL for its polymerase activity using primer extension assays on RPA-coated single-stranded M13 DNA (Fig. 4e). This analysis showed that DNA synthesis by Pol ε^REL is defective relative to Pol ε^wt, resulting in a ~50% reduction of fully extended products at the end of the time course (Fig. 4e). We conclude that POPS is required for efficient Pol ε-mediated strand synthesis.

The reconstituted DNA replication system described above also allowed us to test whether Pol ε^REL directly affects replication initiation. As proficient strand elongation was seen at low levels of Pol ε^REL, total DNA synthesis under these conditions is proportional to the rate of origin firing. Indeed, time course analyses at 15 nM Pol ε showed that overall DNA

synthesis was reduced by up to ~40% in the presence of Pol ε^REL compared with Pol ε^wt, indicating a reduced rate of origin activation (Fig. 4f). In summary, our in vitro results are consistent with in vivo data, supporting a role for POPS in DNA polymerization and replication initiation.

**POPS is important for coping with template barriers.** In addition to reduced DNA polymerization, asymmetric replication structures seen in pol2-REL cells may also be caused by a compromised ability of Pol ε^REL to cope with non-uniformly distributed template blocks. To test this third possibility, we focused on R-loops as an example of template barriers, as R-loop levels can be experimentally modulated via altering R-loop removal enzymes[31]. We thus asked if pol2-REL cells are sensitive to increased levels of R-loops in the absence of the RNA/DNA helicase Sen1 or the nuclease RNase H2[32,33]. Indeed, pol2-REL showed synthetical lethality with either mutant (Fig. 5a and Supplementary Fig. 5a). Though RNase H2 also promotes the excision of misincorporated

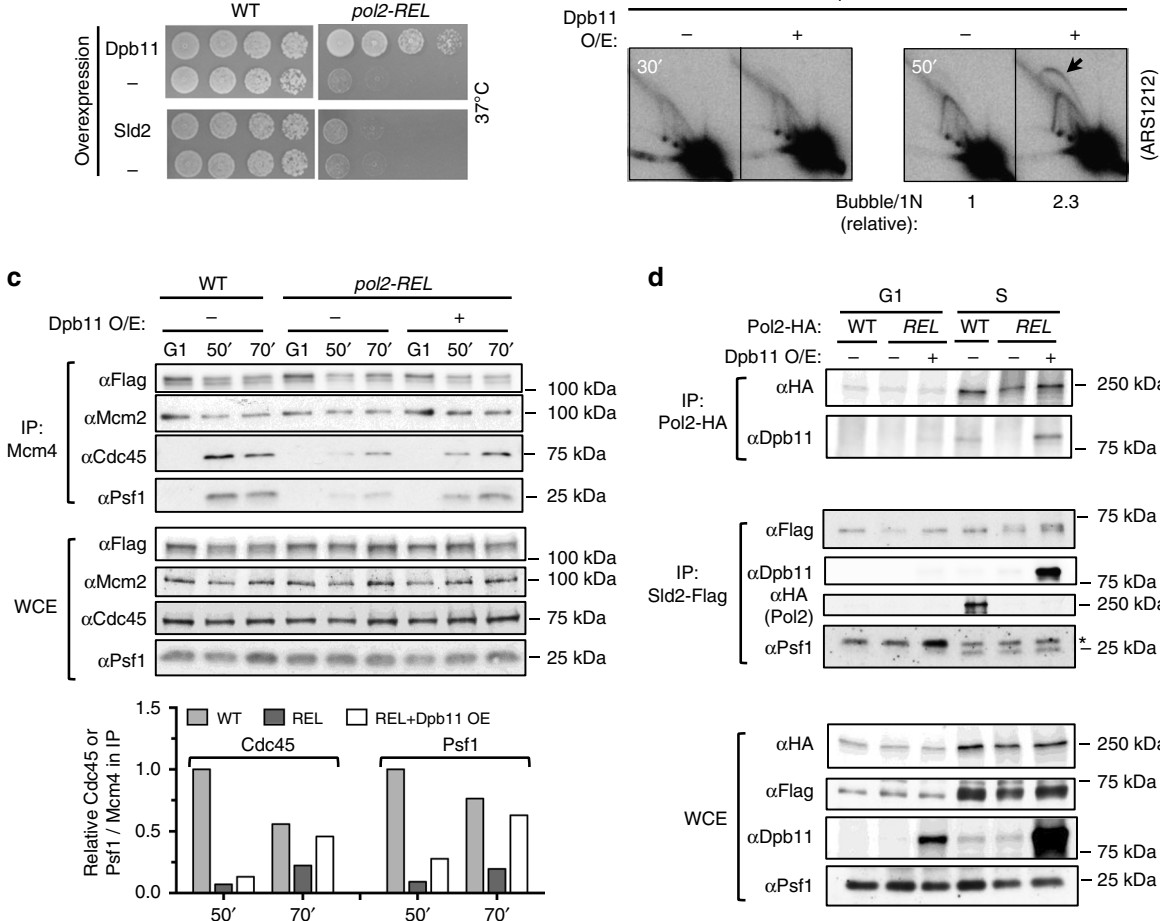

**Fig. 3 Dpb11 overexpression rescues *pol2-REL* defects. a** Increased levels of Dpb11, but not Sld2, rescue *pol2-REL* lethality at 37 °C. Cells with or without (−) a second copy of Dpb11 or Sld2 driven by a galactose inducible promoter were examined on galactose-containing media. **b** Dpb11 overexpression increases replication initiation events in *pol2-REL* cells. Cells growing in medium lacking galactose were arrested in the G1 phase by alpha-factor at 24 °C and then shifted to 37 °C with the addition of galactose to induce Dpb11 overexpression. Releasing cells from G1 into S phase was monitored by FACS analyses (Supplementary Fig. 3b). Two-dimensional (2D) gel data quantification was done and is presented as in Fig. 1e. We note that slow S phase progression in galactose media and high temperature used in this experimental setup yield continuous Y-shaped RI structures rather than large Y-shaped RIs as seen in Fig. 1d. **c** Dpb11 overexpression increases CMG levels in *pol2-REL* cells. Samples were collected as in Supplementary Fig. 3b. Experiments and data presentation are as in Fig. 2a. **d** Increased level of Dpb11 improves its association with *pol2-REL* and Sld2. Cells were collected as in Supplementary Fig. 3b and samples from 120-min time points were examined. Top: HA-tagged wild-type or mutant Pol2 was IP and S-phase specific co-purification of Dpb11 was detected. Middle: Flag-tagged Sld2 was IP, and co-purification of Dpb11, Pol2, and Psf1 was examined. The asterisk indicates the light chain of the antibody used in the IP. Bottom: whole-cell extracts (WCE). For panel c and d, similar results were obtained using at least two independent strains per genotype. Source data are provided as a Source data file.

ribonucleotides from DNA, synthetic lethality between *pol2-REL* and *rnh201Δ* is not related to ribonucleotide excision repair, since a mutant defective in this function (*rnh201-RED*) did not affect *pol2-REL* growth (Fig. 5a and Supplementary Fig. 5a)[34]. We also found that overexpressing another R-loop removal enzyme, RNase H1, partially rescued *pol2-REL* sensitivity to camptothecin, a genotoxin that can increase R-loop levels (Fig. 5b)[31,35,36]. Collectively, these genetic data argue that replisomes containing Pol ε^REL are sensitive to R-loop levels.

A prediction of the above conclusion is that Pol ε^REL would be less proficient at replicating chromosome XII (Chr XII), which is enriched for R-loops and other template obstacles at the ribosomal DNA locus[37–39]. We tested this prediction using pulsed field gel electrophoresis (PFGE) to separate fully replicated chromosomes entering the gel from incompletely replicated branched chromosomes trapped in the wells. This was followed by probing Chr XII

by Southern blotting to derive the ratio of Chr XII signals from gel bands versus signals in the wells. The similarly sized chromosome IV (Chr IV) was used for comparison. Though both chromosomes exhibited delayed replication completion, Chr XII was more strongly affected in *pol2-REL* cells: Chr XII replication in the mutant achieved only 30–60% of wild-type levels at 150–210 min after G1 release, whereas Chr IV replication was about 65–95% of wild-type levels at these time points (Fig. 5c). Thus, *pol2-REL* cells are more severely deficient at replicating chromosomes enriched in template barriers. Since Chr XII contains high levels of other types of barriers in addition to R-loops, our data is compatible with the idea that POPS has a broader role in coping with DNA blockage. In support of this view, removing the Rrm3 helicase that clears protein barriers caused lethality in *pol2-REL* cells (Supplementary Fig. 5a). Our data thus suggest that POPS helps to overcome fork stalling at template barriers.

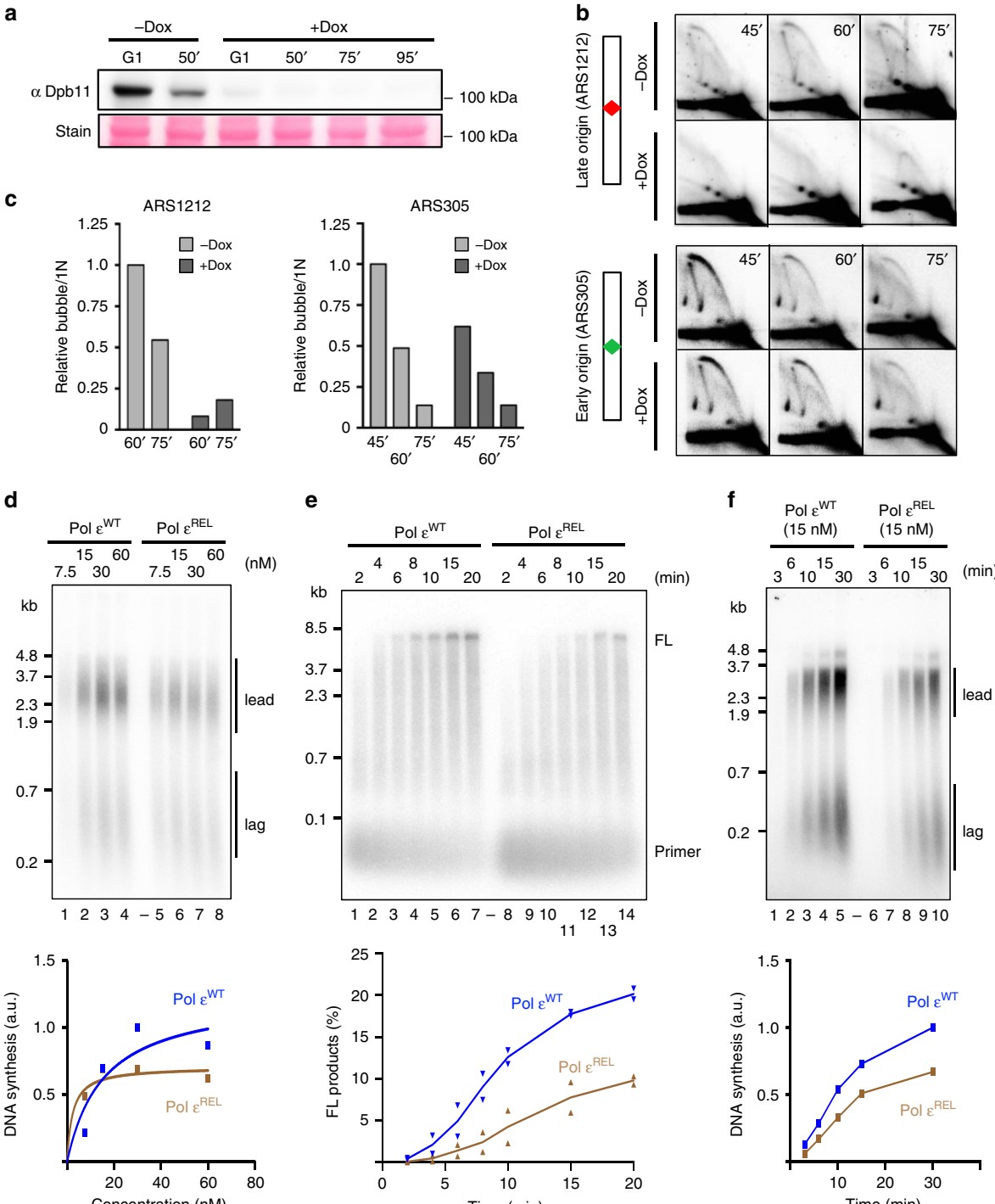

**Fig. 4 Pol ε^REL exhibits reduced activity and replication initiation in vitro. a** Doxycycline addition acutely reduces Dpb11 levels. Western blots verified Dpb11 loss in G1- arrested cells and then cells released into S phase when treated with doxycycline (+Dox), which turned off Tet promoter-driven Dpb11 expression[56]. **b** Dpb11 depletion delays origin firing but does not generate asymmetric replication intermediates. Representative 2D gel pictures are shown. Cells were treated as in panel a; 2D gel analyses were performed and data are presented as in Fig. 1e. **c** Quantification of the bubble-shaped replication intermediates shown in panel b. **d** Reconstituted DNA replication reactions were performed on pARS1 (4.8 kb) in the presence of increasing concentrations of Pol ε^WT (lanes 1–4) or Pol ε^REL (lanes 5–8) as indicated. Replication products were analyzed by alkaline agarose gel-electrophoresis and the total incorporation of ^32P-dATP is plotted (bottom). Lead: leading strands, lag: lagging strands. **e** Primer extension assay with purified Pol ε^WT (lanes 1–7) or Pol ε^REL (lanes 8–14). Single-stranded M13mp18 DNA (7.2 kb) was primed with a radio-labeled oligo, incubated with RFC/PCNA, and reactions initiated by addition of Pol ε and dNTPs. Products were analyzed by alkaline agarose gel-electrophoresis and autoradiography. FL: full length. **f** Time-course analysis of reconstituted DNA replication reactions in the presence of 15 nM Pol ε^WT (lanes 1–5) or Pol ε^REL (lanes 6–10). Replication products were analyzed and quantified as in c. For panel a and b, similar results were obtained using at least two independent strains per genotype. For panel c, d similar results were obtained from two independent experiments. Source data are provided as a Source data file.

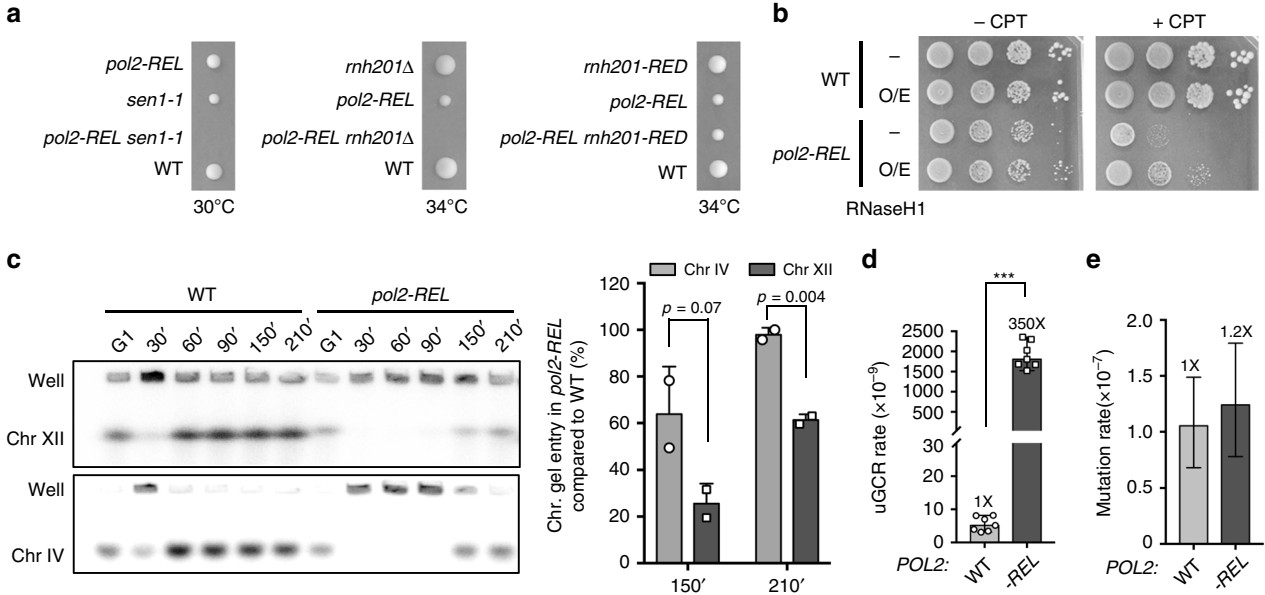

**Fig. 5 pol2-REL cells are sensitive to template barriers and exhibit higher GCR levels but not hyper-mutation. a** pol2-REL is synthetically sick with sen1-1 and rnh201Δ, but not rnh201-RED. Representative tetrads from diploids heterozygous for pol2-REL and tested mutations are shown (more tetrads in Supplementary Fig. 5a). **b** RNase H1 overexpression improves pol2-REL cell growth on CPT-containing media. Cells with (O/E) or without (-) a Gal promoter-driven RNaseH1 construct were spotted at tenfold serial dilution on galactose media containing 2 μg ml⁻¹ CPT. **c** pol2-REL cells exhibit preferential defects in synthesizing Chr XII. Cells were arrested in G1 before release into the cell cycle as in Fig. 1c except that nocodazole was added to prevent the exit from the first cell cycle. Samples were examined by PFGE to trap replicating chromosomes in wells and allow fully replicated ones to enter the gel. Representative results of Southern blots using probes specific for Chr XII or IV are shown (left). Relative ratios of chromosome band signals vs. those in the wells from two independent experiments are plotted and normalized to the wild-type value for 150- and 210-min samples (right). Mean and standard deviation are shown and the statistical difference between mutant and wild-type values was determined by two-tailed Student t-test with p = 0.07 for 150 min and p = 0.004 for 210 min. Similar results were obtained using two independent strains per genotype. **d** pol2-REL cells exhibit higher GCR rates. Two-tailed Mann–Whitney tests were performed for statistical analysis. ***p < 0.001. Error bars indicate 95% confidence intervals. Data are presented as median rates of at least seven cultures from two biological replicates per genotype. **e** pol2-REL does not affect CAN1 forward mutation rate. Error bars indicate 95% confidence intervals. Data are presented as mean rates determined from 72 cultures and two biological replicates per genotype. Source data are provided as a Source data file.

**POPS curbs DNA rearrangements but not hyper-mutation.** We moved on to assess the effects of POPS mutations on genomic stability. We first examined gross chromosomal rearrangements (GCRs). Strikingly, pol2-REL leads to a 350-fold increase in GCR rates at semi-permissive temperatures and a 6-fold increase at the permissive temperature (Fig. 5d and Supplementary Fig. 5b). This is in stark contrast to the lack of mutation rate increases in pol2-REL cells at either temperature as assayed by the loss of the CAN1 gene functions (Fig. 5e and Supplementary Fig. 5c). As a control, we confirmed that the Pol2 EXO mutant pol2-4 showed higher mutation rates (Supplementary Fig. 5c). Our finding is consistent with that POPS mutations being present in non-hyper-mutated cancer cells. In contrast to pol2-REL, pol2-4 showed wild-type levels of GCR rates (Supplementary Fig. 5b). Our findings suggest that different Pol ε functions residing in distinct domains curtail different forms of genomic instability.

The above data suggest that pol2-REL is a separation-of-function allele affecting specific Pol2 roles. To further test this notion, we examined Pol2's role in replication checkpoint activation. Upon treatment with the replication stress agent methyl methanesulfonate (MMS), pol2-REL behaved like wild-type in producing phosphorylated forms of the Rad53 checkpoint kinase and delaying S phase progression, whereas the checkpoint-impaired pol2-11 allele affecting Pol2-CT showed defects in both assays (Supplementary Fig. 5d)[12]. A time-course experiment corroborates the proficiency of pol2-REL in inducing Rad53 phosphorylation in the presence of MMS (Supplementary Fig. 5e). These data support the notion that Pol2-REL maintains the replication checkpoint function.

During normal S phase, pol2-REL also behaved like wild-type, with no detectable Rad53 phosphorylation and no induction of the checkpoint target Rnr4 (Supplementary Fig. 5f). Thus, pol2-REL does not adversely cause checkpoint hyperactivation during S phase. As seen for other replication-defective mutants, pol2-REL cells did show checkpoint activation in G2-M phase, suggesting that its defects are sensed by the checkpoint at this stage of the cell cycle (Supplementary Fig. 5f). Our data suggest that pol2-REL is overall proficient for checkpoint activation and mutation avoidance.

**POPS single mutants perturb replication and increase GCRs.** As each of the examined POPS mutations occurs separately in cancer cells, we queried the consequences of the single-point mutations. When present as single-point mutations, two of the three changes in pol2-REL, R567C, and L621F, recapitulated the pol2-REL phenotype. First, like pol2-REL, pol2-R567C and -L621F were sensitized by mutations in another Pol ε subunit, Dpb2. pol2-REL and pol2-R567C were additionally sensitized by the loss of the Pol δ subunit Pol32 (Fig. 6a and Supplementary Fig. 6a). Second, pol2-R567C and -L621F exhibited severe to moderate delays in bulk genome synthesis (Supplementary Fig. 6b). That pol2-4 did not show these defects further distinguishes Pol2 EXO mutants from POPS mutants (Fig. 6a and Supplementary Fig. 6c). Third, pol2-R567C cells accumulated large Y-shaped DNA molecules, indicating that a single POPS mutation is sufficient to generate asymmetric replication structures (Fig. 6b and Supplementary Fig. 6d). Finally, we found that pol2-R567C cells exhibited a

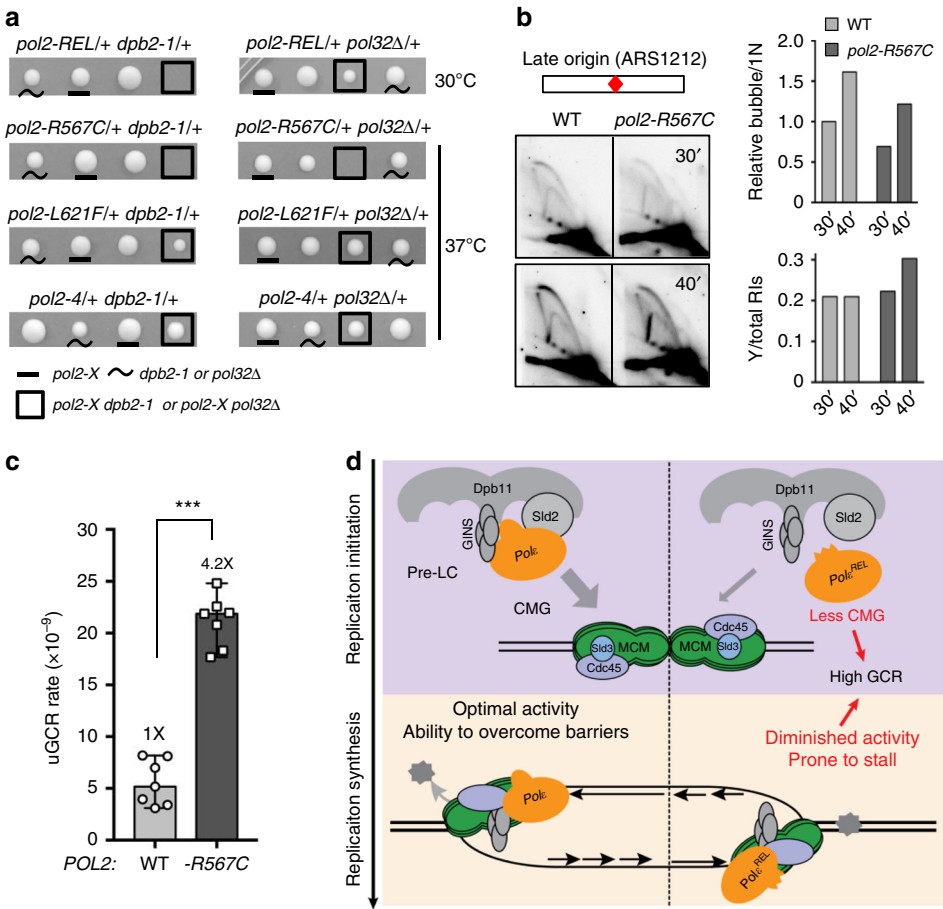

**Fig. 6 Single-point mutations in POPS are sufficient to cause replication defects and GCR increase. a** POPS mutants, but not *pol2-4*, are synthetically sick with *dpb2-1* and *pol32Δ*. Representative tetrads from diploids of indicated genotypes are shown (more tetrads in Supplementary Fig. 6a). **b** *pol2-R567C* leads to reduced replication initiation and progression. Experiments were performed, and data are represented as in Fig. 1e. A represented gel and its quantification are shown. **c** *pol2-R567C* increases GCR rates. Experiments were performed, and data are represented as in Fig. 5d. Two-tailed Mann–Whitney tests were performed for statistical analysis. ***$p < 0.001$. Error bars indicate 95% confidence intervals. Data are presented as median rates of at least seven cultures from two biological replicates per genotype. **d** A proposed model for dynamic roles of the Pol2 catalytic core and its POPS during genome replication. Briefly, our data suggest that POPS enables efficient Pol ε incorporation into the pre-LC and consequently promotes CMG formation and replication initiation; once replication initiates, POPS supports optimal DNA polymerization activity and ability to overcome template barriers. POPS mutations as seen in cancer cells can lead to reduced CMG formation and replication initiation, as well as poor replication progression, resulting in increased genome rearrangements (see text for details).

fourfold increase in GCR rates compared to wild-type cells, suggesting that a single POPS mutation can moderately increase genomic rearrangements (Fig. 6c).

## Discussion

By modeling human POLE cancer-associated mutations in its yeast ortholog, Pol2, we gained insights into native and cancer-associated Pol ε functions. Our data suggest that the Pol2 catalytic core plays a structural role during replisome assembly by enabling Pol ε integration into the pre-LC. Moreover, POPS promotes Pol ε-mediated DNA synthesis and fork passage through template barriers. The distinct effects of POPS at different stages of DNA replication highlight the dynamic functions of the Pol ε catalytic core. Importantly, we showed that perturbation of POPS's functions increases genome rearrangements, thus shedding light on potential genomic consequences of human POLE POPS variants present in cancers.

Genome-wide replication profiles and 2D gel analyses of *pol2-REL* cells suggest decreased replication initiation, particularly at late origins. As *pol2-REL* cells do not adversely turn on the DNA replication checkpoint, a checkpoint-mediated suppression of late origin firing is unlikely. Though we cannot exclude the possibility

that checkpoint activation below the detection limit of our assays might partially contribute to the late origin firing defects in *pol2-REL* cells, our in vitro reconstitution tests showed that Pol ε^REL impairs replication initiation in the absence of checkpoint proteins. Moreover, our biochemical assays found reduced CMG levels in *pol2-REL* cells. These data support a model wherein the Pol2 catalytic core contributes to replication initiation at least in part by promoting CMG assembly (Fig. 6d).

The pre-LC has been proposed to deliver GINS to origin-bound MCM and Cdc45 during CMG formation[28]. However, testing an important prediction of this model, namely that lower pre-LC levels lead to reduced CMG levels, has not been reported. Our demonstration of this point in *pol2-REL* cells fills this gap and provides support for the above theory. As increasing Dpb11 dosage rescues *pol2-REL* cell lethality at non-permissive temperature and restores CMG levels and replication initiation, the effect of POPS on the pre-LC is linked to Dpb11. Considering that Pol2-CT also contributes to pre-LC and replisome assembly and that its mutants' growth defects are rescued by Dpb11 overexpression[40,41], we suggest that Pol2 acts as a central scaffold collaborating with Dpb11 in multiple ways during this important step of replication. That *pol2-REL* cells showed preferential impairment of late origin firing

substantiates the concept that late origins have a reduced capacity for recruitment of replication initiation factors and thus are more sensitive to changes in their levels[29]. This finding adds Pol2 to the growing list of factors that selectively affect genome replication initiation at different loci.

While supporting an important role of the pre-LC in CMG assembly, our data reveal previously unappreciated features and functional flexibility of pre-LC factors in CMG assembly. We found that Dpb11, GINS, and Sld2 can also associate without Pol ε. Given that Dpb11 can bind directly to MCM-associated Sld3, one interpretation of our data is that a "three-member" sub-complex may deliver GINS to MCM via this interaction[42]. The Dpb11 and Sld3 interaction may also allow Pol ε delivery to MCM without forming the pre-LC. Separate means of GINS and Pol ε delivery may be less efficient than the pre-LC, but may still be effective in *pol2-REL* cells. This interpretation could explain that restoration of replication initiation in *pol2-REL* cells by Dpb11 overexpression is associated with enhanced Dpb11-Pol ε interaction but not more pre-LCs. Extrapolating from this idea, inefficient forms of pre-LC variants may be able to account for replisome assembly defects seen in *pol2-NTΔ* mutants[17,43]. Future studies of pre-LC-independent means of CMG assembly will deepen our understanding of pathway plasticity in replisome formation.

Our data show that efficient Pol ε-mediated strand synthesis relies on the integrity of POPS. We found that POPS defects in *pol2-REL* cells lead to asymmetric replication structures and persistent replication intermediates at early and late origins. 2D gel data and genome-wide replication profiles suggest that these structures are not caused by passive replication from adjacent origins, but rather reflect replication fork progression defects. This finding suggests a high degree of uncoupling of the two sister replication forks in *pol2-REL* cells. Although *pol2-REL* cells have low pre-LC levels, we show that such a defect per se does not cause asymmetric replication structures. As POPS is found only in Pol ε among replicative polymerases, our findings reveal an important feature specific for Pol ε-mediated strand synthesis (Fig. 6d).

Our primer extension and reconstituted replication assays show that Pol ε$^{REL}$ exhibited reduced rates of DNA synthesis relative to Pol ε$^{WT}$. In cells, *pol2-REL* leads to a temperature-sensitive phenotype; however, as an elevated temperature inhibits DNA replication in the reconstituted system, we were unable to test Pol ε$^{REL}$ activity at the restrictive temperature in vitro, which may explain the relatively mild, but reproducible, defects observed in this assay. It is noteworthy that a dual effect in replication initiation and elongation caused by Pol ε$^{REL}$ has not been seen after perturbing other replication factors in this experimental set-up, suggesting again a unique requirement for Pol ε at these distinct steps. There can be several possible means by which POPS promotes Pol ε-mediated DNA synthesis, such as facilitating DNA association, promoting dNTP turnover, or increasing processivity. A detailed enzymatic characterization of the Pol ε$^{REL}$ synthesis defect will distinguish among these possibilities in the future. In addition, it will be interesting to determine how POPS functions are regulated by sumoylation, which was recently reported to occur in this region[44,45].

Our data also suggests an increased propensity of Pol ε$^{REL}$ replisomes to stall at template barriers. A special requirement for POPS by Pol ε likely reflects a unique challenge for synthesizing long stretches of DNA during leading strand elongation, which requires high processivity and the ability to resume synthesis upon encountering many template barriers. One way that Pol2 family proteins acquire these attributes is likely through obtaining additional domains. The P-domain within the Pol2 catalytic core increases Pol ε processivity by binding to template DNA[18].

Further studies will be needed to elucidate the underlying mechanisms by which POPS can promote Pol ε activity and its interplay with the P-domain.

We found that Pol2-REL largely maintains functions in mutation avoidance and checkpoint activation (Fig. 6d). Though inhibition of origin firing has been linked to checkpoint activation, it is conceivable that the mutant's influence on origin firing in normal growth condition is not the major contributing factor for Pol2-mediated checkpoint activation in MMS conditions, which requires Pol2-CT function. Our data thus suggest that *pol2-REL* supports a phenotype distinct from previously studied Pol2 mutants[43,46].

We show that an intact POPS is required for curbing genomic rearrangements but not mutation rates, demonstrating that it affects genomic instability in a manner distinct from the Pol2 EXO domain (Fig. 6d). Given the conserved nature of POPS, human POPS may confer tumor-suppressive effects without affect mutation rates. This theory is consistent with the recent suggestion that POLE defects contribute to non-hypermutagenic tumors[24]. It is noteworthy that the *pol2-12* checkpoint defective allele also leads to increased GCR levels[47]. As the pathogenic significance of the majority of non-EXO cancer-associated POLE mutations remains unknown, our work suggests that modeling these mutations in yeast can provide an effective way to distinguish their genomic instability effects between DNA hyper-rearrangements versus hyper-mutations.

Among cancer-associated mutations examined, *pol2-R567C* exhibited the strongest defects, while *pol2-L621F* showed a milder phenotype and *pol2-E611K* had largely normal behavior. This finding differs from a computational assessment of mutational effects, which often assume that the most drastic changes of the chemical properties of conserved residue would be the most disruptive to protein function. Our findings thus highlight the value of yeast studies to reveal distinct in vivo effects of point mutations found in cancer cells that cannot be accurately predicted by in silico analyses. We note that *pol2-R567C* showed less severe defects compared with *pol2-REL*, however, it may still be sufficient to cause accumulative genome changes that influence cell proliferation. Our work also suggest chemicals and conditions that sensitize POPS mutants in yeast. This may inform the design of selective killing of cancer cells harboring POPS mutations.

## Methods

**Yeast strains and genetic manipulation.** Yeast strains are listed in Supplementary Table 1 and are derivatives of W1588-4C, a RAD5 variant of W303 (*MAT*a *ade2-1 can1-100 ura3-1 his3-11,15 leu2-3, 112 trp1-1 rad5-535*)[48]. At least two strains per genotype were examined in each experiment, and only one is listed for each genotype in Supplementary Table 1. Standard PCR-based methods were used to generate integrated alleles and add tags to proteins at endogenous loci, followed by DNA sequencing verification. Plasmids are listed in Supplementary Table 2. Standard molecular methods were used for cloning, followed by sequence verification. The preparation of media was according to standard procedures. Genotyping in tetrad analyses (Figs. 5a, 6a and Supplementary Figs. 5a and 6a) was done as followings: (i) standard drug selection or autotrophic marker selection was used to score POPS mutations (marked by *KAN*), *pol32Δ* (marked by *HYG*), and *sen1-1* (marked by *HIS3*), (ii) *dpb2-1*, *pol2-4* and *rnh201-RED* were scored by PCR combined with specific restriction digestion using SfcI, SfcI and ApaI restriction enzymes, respectively, and (iii) *rhn201Δ* and *rrm3Δ* were scored by PCR primers flanking the gene. Primer sequences are listed in Supplementary Table 3.

**Cell cycle arrest and release.** G1 arrest of yeast culture was achieved using a standard protocol. Briefly, log-phase cultures were treated with alpha-factor (5 μg ml$^{-1}$) until >90% of cells exhibited G1 arrest. For experiments involving temperature shift, G1-arrested cells were shifted to 37 °C for 1 h. 300 μg ml$^{-1}$ pronase (Millipore) was then added into the G1-arrested culture to release the cells. Samples were collected at multiple time points for examination. For experiments involving galactose induction, 2% Raffinose was used in growing log-phase culture and 2% galactose was added to induce gene expression. Flow cytometry analyses were performed using a standard procedure. Briefly, ethanol fixed yeast cells were washed with and resuspended in sodium citrate solution. RNase and Proteinase K

were added sequentially to remove RNAs and proteins. Sytox green was then used to stain DNA. Flow cytometry was performed using BD LSRII flow cytometer, and data were analyzed with the FlowJo software. Gating strategy is described in Supplementary Fig. 7.

**Two-dimensional (2D) agarose gel electrophoresis.** Two-dimensional (2D) gel analyses were performed using a standard protocol[49]. Yeast cells were treated with zymolyase to produce spheroplasts. After cell lysis and proteinase K treatment to degrade proteins, DNA was purified by CsCl gradient centrifugation and precipitation. Extracted DNA was digested by *Eco*RI and separated by agarose gel electrophoresis in two dimensions. DNA was then transferred onto Hybond-XL membranes (GE Healthcare) and analyzed by Southern blot using probes hybridizing specifically to ARS305, ARS1212, or ARS315. Primers used for probe amplification are listed in Supplementary Table 3. For quantification, the signals of 1N DNA were obtained from shorter exposures, while those of DNA intermediates came from longer exposures to ensure both types of signals fell within the linear range of detection on the PhosphorImager.

**PFGE analysis.** PFGE was performed using a standard protocol[50]. Briefly, cells were embedded in agarose plugs, spheroplasted, and deproteinized. Plugs were loaded on 0.5X TBE gels and run on a Bio-Rad CHEF-DR III PFGE System for 15 h to achieve chromosome separation. Chromosomes were transferred onto Hybond-XL membranes (GE Healthcare) using standard capillary transfer technique, and membranes were analyzed by Southern blot using probes hybridizing specifically to chromosome XII or IV. Primers used for probe amplification are listed in Supplementary Table 3. We note that the same number of cells were examined in each synchronized S phase timepoint, and consistent loading is evident from the similar amount of Chr XII and IV in wild-type cells in Fig. 5c. As the chromosome replication was assessed by the signal ratio of chromosome entering the gel (completed replication) vs. that retained in the well (incomplete replication), results are not affected by small loading differences.

**Whole-genome sequencing and copy number calculation.** Genome sequencing and copy number calculation were carried out using a standard protocol[26]. Briefly, wild-type and *pol2-REL* cells were collected at G1 and S phase at 24 °C. In all, 1.5 µg genomic DNA from each sample was used to generate libraries with a KAPA library kit (iGO facility, Memorial Sloan Kettering Cancer Center) and sequenced with a HiSeq 2500 (Illumina). At least 10 million 50-bp paired-end reads were generated per sample. Reads were mapped to S288c reference genome (SGD, SacCer2) and summed into 1 kb bins using Genome Brower after excluding repetitive sequences. The binned reads from the S phase sample at a given locus were divided by those from the G1 sample and normalized to the ratio of total reads to give a genome-wide mean value of 1. The number was adjusted by the relative DNA content in the S phase derived from the FACS fitting curve (Fig. 1c) to the relative copy number of a particular locus. The maps of adjusted copy numbers were smoothed with the LOESS function. For the meta-analysis of DNA origins, the copy numbers of the DNA sequences spanning 20 kb upstream and downstream of both early origins and late origins were averaged[49]. Un-replicated regions ("flag regions" on the copy number plot) on both sides of the origins was set as the baseline (copy number = 1).

**Co-immunoprecipitation and western blots.** G1 and S phase cells were harvested. Note that a mild cross-linking step was included before cell harvesting for examining pre-LC formation using a standard protocol[28]. Briefly, cultures were incubated with 1% formaldehyde for 20 min before quenching with 120 mM glycine for 5 min. Cells were disrupted by glass bead beating in lysis buffer, and Benzonase was added to digest nucleic acid before centrifugation for 30 min at 20,000 x g to obtain whole-cell extract (WCE). The lysis buffer used for examining pre-LC formation included 50 mM HEPES-KOH, pH 7.5, 140 mM NaCl, 1% TritonX-100, 2 mM MgCl₂ and cOmplete™ Ultra EDTA free protease inhibitor (Roche). The lysis buffer used in other experiments included 50 mM HEPES-KOH pH 7.5, 100 mM KOAc 1% TritonX-100, 2 mM MgOAc, 2 mM NaF, 2 mM beta-glycerophosphate, 10 mM beta-mercaptoethanol, and cOmplete™ Ultra EDTA free protease inhibitor (Roche). WCE was incubated with pre-washed beads, including anti-flag beads (A2220, Sigma-Aldrich), anti-HA beads (ThermoFisher Scientific), or IgG Sepharose 6 Fast Flow (GE Healthcare) for 2 h at 4 °C. For Fig. 2b top panel and Fig. 3d middle panel, Flag or HA antibody plus Protein G beads were used for IP. After washing the beads, bead-bound proteins were eluted with 2x Laemmli buffer without DTT or elution buffer (50 mM Tris-HCl [pH 8.0], 10 mM EDTA, 1% SDS). Proteins were boiled for 5 min before subjected to SDS-PAGE on 4-20% gradient gels (Bio-Rad) and transferred to nitrocellulose membrane (GE healthcare) for Western blotting. Antibodies used in probing western blots include anti-Dpb11 and anti-Cdc45 (gift from B. Stillman), anti-Psf1 (gift from K. Labib), anti-Dpb2 (gift from H. Araki), anti-Mcm2 (Santa Cruz, sc-6680), anti-Rad53 (Abcam, ab104232), anti-Flag (Sigma-Aldrich, F1804), anti-Myc (Bio X Cell, BE0238), anti-HA (Roche, 11867423001), Peroxidase Anti-Peroxidase (Sigma-Aldrich, P1291), anti-Rnr4 (Abcam, anti-Tubulin, ab6160). Uncropped scans of all blots with at least one molecular weight marker labeled are included in Source data file.

**GCR and mutation rate assays.** GCR assays were performed using a standard protocol and rates were calculated[51,52]. To ensure all experiments are in the same genetic background, we moved the GCR assay to W303 background[53]. For each genotype, at least seven cultures were examined. Cells were plated on SC + 5-FOA + Can (FC) and SC plates to obtain colony numbers that lose the URA3-CAN1 cassette and total viable colonies, respectively. GCR rates were calculated as m/NT, wherein $m$ $(1.24 + \ln[m]) - NFC = 0$. $m$: mutational events, NFC: number of colonies on FC plates, NT: colonies formed on SC plates. The upper and lower 95% confidence intervals were then derived. The URA3-CAN1 cassette is inserted at YEL072w and YEL068c in dGCR and uGCR assay, respectively. Mutation rates were determined by fluctuation assays[54,55]. Briefly, dilution of a saturated overnight culture was used to determine cell number per culture and was plated on media containing canavanine. Mutants were counted after two days of growth and mutation rates were calculated using the Ma–Sandri–Sarkar maximum likelihood method and 95% confidence intervals were calculated.

**Primer extension assays.** RPA, RFC, PCNA, and Pol ε were purified using a standard protocol[30]. Reactions were performed at 30 °C in polymerization buffer (25 mM Hepes-KOH pH 7.6, 10 mM Mg-Acetate, 0.02% NP-40S, 60 mM K-Acetate, 5% glycerol). The reactions contained 1 nM circular DNA template (M13mp18 ssDNA, NEB) annealed to a radio-labeled oligo ($^{32}$P-5′-CCCAGTC ACGACGTTGTAAAACG), 60 nM PCNA, 10 nM RFC, 400 nM RPA, 5 mM ATP, 2.5 mM DTT, 0.1 mg ml$^{-1}$ BSA and 100 nM of either Pol ε$^{WT}$ or Pol ε$^{REL}$. Polymerase reactions were initiated by the addition of dNTPs (200 µM each) to the mix. Reactions were stopped at indicated times by taking an aliquot and adding 50 mM EDTA and 0.4% SDS. Products from the reaction were separated on 0.8% alkaline agarose gel (30 mM NaOH and 2 mM EDTA), dried, and imaged using Typhoon FLA 7000. Quantification of the gel images was performed using the ImageJ software.

**DNA replication assays.** Reactions were carried out using pARS1 (4.8 kb) as template and proteins were purified as shown previously[30]. All the steps of the replication assay were carried out at 30 °C. First, Mcm2-7 loading was performed in a 10 µl reaction by incubating 10 nM plasmid DNA template, 50 nM ORC, 50 nM Cdc6 and 100 nM Cdt1·Mcm2-7 in a buffer consisting of 25 mM HEPES-KOH (pH 7.6), 0.02% NP-40, 10 mM magnesium acetate, 5% glycerol, 100 mM potassium acetate, 2 mM DTT and 5 mM ATP for 15 min. DDK was then added to 125 nM and incubation continued for a further 15 min. DNA replication was initiated by the addition of 15 µl master mix of replication proteins resulting in final concentration of 60 nM Sld3·7, 80 nM Cdc45, 50 nM CDK, 80 nM GINS, 30 nM Dpb11, 80 nM Sld2, 120 nM RPA, 60 nM Pol α, 35 nM Ctf4, 20 nM RFC, 70 nM PCNA, 4 nM Pol δ, 20 nM Csm3·Tof1, 20 nM Mrc1, 15 nM Mcm10, 30 nM Top1, and 30 nM Top2. Concentration of Pol ε$^{WT}$ or Pol ε$^{REL}$ was 30 nM in Fig. 4e, or as indicated in Fig. 4f. The final replication reaction also included 0.2 mg ml$^{-1}$ BSA, 40 µM each dATP/dGTP/dTTP/dCTP, 200 µM each GTP/CTP/UTP, 66 nM α$^{32}$P-dATP (3000 Ci mmol$^{-1}$), 16 mM creatine phosphate, 0.04 mg ml$^{-1}$ creatine kinase, 190 nM potassium acetate, 20 mM sodium chloride, and 15 mM potassium chloride. Reactions were terminated after 30 min by adding 40 mM EDTA, 1.6 U Proteinase K and 0.3% SDS and incubating the mix at 37 °C for 30 min. DNA was isolated by phenol/chloroform extraction and unincorporated nucleotides were removed with G-50 spin columns equilibrated with TE buffer. The sample was then fractionated on a 0.8% alkaline agarose gel (30 mM NaOH and 2 mM EDTA), dried, and imaged using Typhoon FLA 7000. Quantification of the gel images was performed using the ImageJ software.

**Protein purification.** Proteins used in the replication and primer extension assays were purified to close to homogeneity. The purification scheme was modified for the following proteins as described below. In all cases, cells carrying galactose inducible protein expression constructs were grown at 30 °C in YP-GL (YP + 2% glycerol/2% lactic acid) to a density of $2–4 \times 10^7$ cells ml$^{-1}$. Galactose was then added to 2% and cell growth continued for 4 h. In the case of Dpb11, cells were arrested in G1 phase for 3 h by addition of 100 ng ml$^{-1}$ alpha-factor prior to the addition of galactose. Cells were harvested by centrifugation and washed once with 1 M sorbitol/25 mM Hepes-KOH pH 7.6 followed by a second wash with buffers as indicated. Washed cells were resuspended in 0.5 volumes of respective buffers as indicated and frozen dropwise in liquid nitrogen; the resulting popcorn was stored at −80 °C. Frozen popcorn was crushed in a freezer mill (SPEX CertiPrep 6850 Freezer/Mill) for six cycles of 2 min at a rate of 15 impacts per second. Extracts were clarified by centrifugation at 195,000 x g for 45 min (T647.5 rotor), and subject to different types of purification scheme as described. All proteins were stored at −80 °C in aliquots.

To purify Csm3 and Tof1 proteins, cells were washed with buffer C (25 mM Hepes-KOH 7.6/0.02% NP-40 S/10% glycerol/1 mM DTT)/100 mM NaCl and resuspended in ½ volume of buffer C/100 mM NaCl/protease inhibitors for preparation of cell popcorn. Cell powder was thawed on ice, one volume of buffer C/100 mM NaCl was added, and the resulting suspension supplemented with an additional 200 mM NaCl prior to clarification of the extract by ultracentrifugation. The soluble extract was supplemented with 2 mM CaCl₂ and incubated with calmodulin affinity beads for 3 h at 4 °C. The beads were washed with 10 CV

(column volume) buffer C/300 mM NaCl/2 mM CaCl$_2$, and bead-bound protein eluted with 6 CV buffer C/300 mM NaCl/1 mM EDTA/2 mM EGTA. Eluates were pooled, diluted with an equal volume of buffer C, and fractionated on a Mono Q 5/50 GL column using a gradient of 150–1000 mM NaCl over 30 CV. Peak fractions from the Mono Q step were pooled and run through Superdex 200 gel filtration column equilibrated in buffer C/300 mM KOA and collect the peak fractions.

To purify the Mrc1 protein, cells were washed with buffer M (25 mM Hepes-KOH pH 7.6/1 mM EGTA/1 mM EDTA/0.02% NP-40 S/10% glycerol/0.5 mM DTT)/100 mM NaCl and resuspended in ½ volumes of buffer M/100 mM NaCl/protease inhibitors for preparation of popcorn. Cell powder was thawed on ice, one volume of buffer M/100 mM NaCl was added, and the resulting suspension was supplemented with an additional 300 mM NaCl prior to clarification of the extract by ultracentrifugation. The clarified extract was incubated with M2-agarose anti-FLAG beads at 4 °C for 3 h. The beads were washed with 10 CV of buffer M/400 mM NaCl, resuspended in buffer M/400 mM NaCl/10 mM Mg-Acetate/1 mM ATP and incubated for 10 min at 4 °C, washed again with 10 CV of buffer M/400 mM NaCl, and bound protein eluted with 1 CV of buffer M/400 mM NaCl/FLAG peptide (0.5 mg ml$^{-1}$) followed by 2 CV of buffer M/400 mM NaCl/FLAG peptide (0.25 mg ml$^{-1}$). Eluates were pooled and diluted with equal volume of buffer M and fractionated on Mono Q column with a salt gradient of 200–1000 mM NaCl over 10 CV. Peak fractions from Mono Q step were pooled and dialyzed against 25 mM Hepes-KOH pH 7.6/1 mM EDTA/0.02% NP-40-S/40% glycerol/150 mM NaCl/0.5 mM DTT.

To purify the Dpb11 protein, cells were washed with buffer D (45 mM Hepes-KOH pH 7.6/0.02% NP-40 S/10% glycerol/1 mM DTT)/100 mM NaCl and resuspended in ½ volumes of buffer D/100 mM NaCl/protease inhibitors for popcorn preparation. Cell powder was thawed on ice, one volume of buffer D/100 mM NaCl was added, the resulting suspension supplemented with an additional 200 mM NaCl and 0.45% polymin P pH 7.3, and the extract clarified by ultracentrifugation. The clarified extract was supplemented with 2 mM CaCl$_2$ and incubated with calmodulin affinity beads for 6 h at 4 °C. The beads were washed with 10 CV buffer D/300 mM NaCl/2 mM CaCl$_2$, and the bead-bound protein eluted with 10 CV buffer D/300 mM NaCl/1 mM EDTA/2 mM EGTA. Peak fractions were pooled, dialyzed against buffer D/150 mM NaCl, and fractionated on a Mono Q 5/50 GL column with a salt gradient of 150–1000 mM NaCl over 20 CV. Peak fractions from the Mono Q step were pooled, spin-concentrated, and fractionated on a Superdex 200 gel filtration column equilibrated in buffer D/300 mM KOAc to collect the peak fractions.

To purify the Sld2 protein, cells were washed with buffer S (25 mM Hepes-KOH pH 7.6/0.02% NP-40 S/10% glycerol/1 mM EDTA/1 mM DTT)/100 mM KCl and resuspended in ½ volumes of buffer S/100 mM KCl/protease inhibitors for popcorn preparation. Crushed cell powder was thawed on ice, one volume of buffer S/100 mM KCl was added, and the resulting suspension was supplemented with 400 mM KCl prior to clarification of the extract by ultracentrifugation. The clarified extract was incubated with M2-agarose anti-FLAG beads for 1 h at 4 °C. The beads were washed with 10 CV buffer S/500 mM KCl and bead-bound protein eluted with 1 CV of buffer S/500 mM KCl/FLAG peptide (0.5 mg ml$^{-1}$) followed by 2 CV of buffer S/500 mM KCl/FLAG peptide (0.25 mg ml$^{-1}$). Peak fractions were pooled, dialyzed against buffer S/350 mM KCl and stored in aliquots. Pol ε$^{WT}$ was purified as described previously[30] but from asynchronous cells. Pol ε$^{REL}$ was purified identically to Pol ε$^{WT}$.

**Reporting summary**. Further information on research design is available in the Nature Research Reporting Summary linked to this article.

## Data availability

The whole-genome-sequencing data associated with this study are available through GEO accession GSE132450. The source data underlying Figs. 1b, e, 2a–c, 3a–d, 4a–f, 5b–e, 6b, c and Supplementary Figs. 1b, c, 2c–f, 3a, c, 5b–f, 6d are provided as a Source data file. All data is available from the authors upon reasonable request.

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

## Acknowledgements

We thank H. Araki, K. Labib, and B. Stillman for sharing strains and antibodies, X. Peng, J. Bonner, P. Sarangi and other Zhao lab members for editorial comments. This work was supported by a Functional Genomics Initiative Pilot Award and grants from the National Institute of General Medical Science (GM080670 and GM131058) to X.Z. and a grant from the National Institute of General Medical Science (GM107239) to D.R.

## Author contributions

X.M., L.W., S.D., D.R., and X.L. designed experiments. X.M. and L.W. performed all in vivo experiments and S.D. performed in vitro replication assays. T.Z. and J.X. analyzed the sequencing data. X.M. and X.Z. wrote the manuscript with review and editing from L.W., S.D., D.R., T.Z., and J.X.

## Competing interests

The authors declare no competing interests.
