## [Peer Review File · Nature Communications]

Reviewers' comments:

Reviewer #1 (Remarks to the Author):

Review of Meng et al (Remus and Zhao) "The Pol epsilon catalytic core plays a structural role in replisome assembly and relies on a unique domain for efficient strand synthesis"

Summary: The study is an exceptional example of how model organisms can be used to explore disease-associated mutations in conserved proteins required for core cellular processes to advance basic research as well as illuminate the underlying mechanisms of disease. This study deserves publication in *Nature Communications*.

This manuscript describes an impressively comprehensive approach to define a novel functional region of the core catalytic subunit in the major eukaryotic DNA polymerase, Pol ϵ . The core catalytic subunit, Pol2 in yeast and POLE in humans, is responsible for leading strand synthesis and has structural and functional features that clearly distinguish it from other DNAPols, but how these distinct structural features confer this polymerases unique functions in biology is unclear. Mutations in POLE are associated with and likely function as major drivers of some cancers. Understanding the fundamental role of Pol ϵ is thus of critical importance to both basic and health-related research advances.

The basic approach was to model specific cancer-associated mutations found in the human ortholog POLE into the yeast ortholog Pol2 and use the experimental power of the yeast model to define the role for this conserved Pol2 region. While this disease-motivated reverse-engineering approach is becoming more standard, this study did an exceptional job and could serve as a primer for how to do this kind of disease-modeling study well. The interesting and sound rationale, the multitude of experimental approaches, and the extremely high quality of the data make this study stand out. The authors focus on non-EXO POLE mutations (whose mechanism and etiology are well studied) found in some cancers that affect a unique conserved region peripheral to the catalytic core. They refer to this region as POPs. POPs is interesting for two major reasons: 1. It is specific to Pol ϵ but its functions are completely unknown; 2. Mutations in this region are associated with specific types cancers, but because there is no mechanistic understanding of this Pol2 region there is no clue as to how these mutations cause cellular defects that promote these cancers.

The authors use extremely sound genetic reasoning and experimentation to engineer yeast where the a Pol2 gene (the catalytic subunit of interest in POLE in humans and Pol2 in yeast) is replaced with a mutant version that encodes three distinct cancer-associated amino acid substitutions, pol2-REL, and show that pol2-REL yeast are temperature-sensitive. (Later they show that single amino acid substitutions, as they are found in human cancers, recapitulate key phenotypes of pol2-REL). These pol2-REL yeast are then used as a tool in subsequent experiments to elucidate how POPs contributes to normal Pol ϵ functions. pol2-REL produced normal levels of Pol2 yet showed numerous novel replication defects, indicating that POPs was important for normal Pol ϵ function and defined a new mechanistic role(s) for Pol2 in genome stability.

A major strength of the study was that multiple independent experiments were used to reinforce important interpretations. For example, direct high-resolution origin mapping and co-IP experiments, in vitro studies, and genetic interaction analyses, each generating beautiful robust data, supported the model that POPs was critical for promoting CMG formation (active form of the helicase) and efficient replication fork progression. These observations in turn were fully consistent with POPs promoting genome stability by suppressing GCRs rather than mutation rate in yeast, and this observation offered an intriguing and reasonable explanation for the etiology of cancers associated with defects in this domain. The basic and disease-relevant aspects of the research

were highly complementary.

While the rationale and experimental execution of the study were outstanding, there were some issues of the manuscript that could be strengthened:

In some sections the writing was dense, with more emphasis placed on what seemed like relatively minor points than ideal. These issues detracted from the high impact results. For example, there is no need to posit why one of the single amino acid substitutions did not recapitulate the pol2-REL phenotype. What is remarkable is that two did and that the defects make total sense with the new function in Pol2 posited in the model.

The title could better emphasize the high impact nature of the work. The title should highlight how the model organism illuminated the pathology of disease-linked mutations to illuminate both basic and disease-relevant mechanisms.

When possible, the genetics conclusions should be stated in a way to enhance comprehension. For example, consistently use positive statements, i.e. line 26 heading, e.g. "POPS promotes replication initiation and prevents asymmetric fork progression" really means simply "The POPS domain promotes both replication initiation and fork progression". The asymmetric replication structures are merely the data you use to conclude that there is a defect in RF progression from my understanding.

Emphasize that you observe an accumulation of large Y-structures but not small Y-structures. Small Y's would indicate passive replication from a neighboring origin, but you do not see those accumulating. The simplest (only?) explanation is that the two RFs on the fragment are not progressing with equal efficiency, so you see an accumulation of large forks.

Is it possible given your data that at origins, that only a single CMG is being established, and so at a given origin in a population, those large Y's are a result of only single directional RF forming at an origin (that is, the origin is no longer an origin of bidirectional replication)? It seems this could also result from reduced CMG formation. So the molecular defect is not necessarily always/only because of direct defects in RF progression caused by pol2-REL defective Pol ϵ but rather due to only a single CMG forming in the first place (which of course is ultimately due to the pol2-REL defective Pol ϵ)? If this interpretation is fair, then there are possible two distinct mechanisms by which Pol2 promotes RF progression--failure to form enough CMGs at initiation and unstable replisomes that have trouble tolerating RF barriers? (Is that what is being conveyed in the Model Fig 6D?)

While no hyper-activation of checkpoints could be detected in pol2-REL cells, isn't it difficult to conclude with certainty that the greater defects observed at Late origins are not due to checkpoint signals, below detection, from the many problems that you show are clearly occurring during replication of earlier regions of the genome in pol2-REL cells?

The conclusions about the pre-LC defects versus the pre-LC independent defects (i.e. that Dpb11-Pol2 (via POPS) was somewhat confusing and hard to follow. Are you saying that the primary/relevant defect of pol2-REL is not that Pol2 is having trouble joining the pre-LC per se but that its defect interacting with Dpb11 itself is the problem that leads to reduced CMGs and reduced RF progression? It was confusing in how it was highlighted after spending time introducing the pre-LC defects as a possible mechanism.

Reviewer #2 (Remarks to the Author):

The manuscript submitted analyses the function of a region of Pol2, the catalytic subunit of DNA Pol ϵ , named by the authors POPS. By introducing some mutations observed cancer, the author suggests that this region plays a role in:

- Origin firing
- Pol ϵ catalytic activity
- Symmetry of fork progression

The authors indicate that the origin firing defects and the defects in fork progression are 2 independent processes that depend on the integrity of POPS. These mutations lead to a sizable increase in gross chromosomal rearrangement, providing an explanation for the chromosomal instability observed in some cancers.

This work is interesting and the phenotypes are clearly and convincingly presented. Moreover, these data suggest a novel understanding of the roles of the catalytic domain of Pol2 during origin firing. I think this paper would be of great interest for people in the field. Since the novelty of the model suggested in the paper, however, I believe that more work is required to corroborate the mechanistic model put forward by the authors, thus to support the claims made in the discussion. For this reason I would suggest:

Major points:

This paper reaches different conclusions from the ones in Muramatsu et al., 2010. I believe this need to be addressed. Araki's group shows, using convincing in vitro experiments, that Pol ϵ can bind individually to Dbp11, GINS and Sld2 (especially phospho-Sld2). Also, Dpb2 binds directly GINS (as seen also in Sengupta et al., 2013). Is this the case also in pol2-REL (in Fig 2B)? Moreover, GINS binding to Dpb11 and phospho-Sld2 depends on Pol ϵ in Muramatsu et al., 2010. The author suggests instead that Dpb11-Sld2 interact with Psf1 independently of Pol ϵ - although in a non-stoichiometric manner (Fig 3D, S3C) -. The author should address this difference and prove the existence of a Sld2-Dpb11-GINS subcomplex. These data would be more convincing if the author could recapitulate and confirm this model with purified proteins.

Fork asymmetry is among the most striking and interesting aspects of the paper. The authors indicate that this does not depend on the defects in origin firing, since depletion of Dpb11 does not show fork asymmetry. On the other hand, DPB11 overexpression suggests a loss of fork asymmetry in pol2-REL (Fig 3B)? Or is this because the experiment is at 37°C? If so, what would happen at 24°C? It would be also interesting to understand where this stalling occurs at ARS 305 and 1212. Could the author use different restriction enzymes so that the replication origin is no longer at the mid-point of a restriction fragment? Are there any specific features that might explain the stalling (e.g. transcription termination)?

The authors clearly show decreased DNA synthesis in Pol ϵ REL. Is this caused by a decrease in catalytic activity or lower association of Pol ϵ REL to the DNA/replisome and/or an increased turnover?

The link between the levels of origin firing and checkpoint activation dynamics has been long established, as seen for example in mutants of SLD2 (Wang H, Elledge SJ 1999) or ORC (Shimada et al., 2002). Moreover, the function of Pol2 in checkpoint activation is known (Navas TA, et al, 1996). The authors claim that pol2-REL has defects in origin firing but not checkpoint defects. I would suggest to check whether the dynamics of Rad53 activation are affected in a time course, in strains +/- RAD9.

Minor points:

The biochemical experiments are conducted using different conditions (permissive and non-permissive temperatures). This makes somewhat confusing the analysis of the results and the comparison between different experiments. The authors see severe pre-IC defects at 24°C, but a more limited impact on replisome formation (S2E). DPB11 overexpression partially suppress origin firing at 37°C (Fig 3C, S3B) but not pre-IC (120' 37°C). What happens at 24°C (so to easily compare to 2A, B) and in a wild type O/E DPB11? Similar conditions might help comparing results.

In Fig 1D, the early origins, even in wild type, appear to be not really efficient (about 20% fully duplicated). Is my interpretation of the data correct? Why does this look quantitatively different to the data in S1D?

Is the binding of Pol ε to the CMG affected in pol2-REL? Is this the reason why the mutation is synthetic lethal with dpb2-11?

Fig 2 D: not sure quantifications reflect what is shown; in wild type the largest amount of Cdc45/Psf1 co-IPing with Mcm4 is at 30' in the film but not in the quantification. The amount of Mcm4, however, appears constant.

I struggle to understand how R-loops (in front of the CMG) might be specifically toxic in a mutant of POL2 that is catalytic deficient. Alternatively, any barrier to fork progression might be a challenge in pol2-REL mutants. What happens, for example, in a rrm3Δ background?

Really minor

Line 78-79: sentence not clear (is it the P domain?)

line 172: reference repeated

How did the authors decided to use the mutation S595P? this does not follow the rational of the rest of the mutations used.

Colour of the crystal structure makes hard to see the POPS region.

Psf1 shows an aspecific band in some IPs but not others. An untagged control help making clear which one is the background band.

Reviewer #3 (Remarks to the Author):

Meng et al

This is an interesting, mostly well-done study in which the authors examine the role of a small structural domain of PolE/Pol2 in the function of DNA PolE. Through the use of a large number of different assays and an imaginative use of cancer-associated mutations, the authors provide strong evidence for roles of this domain in replisome assembly and DNA synthesis as well as the importance of this domain in vivo. Overall, it is a very solid study that is a worthy contribution to the field.

I do have some concerns that need to be addressed. The results on the cancer-associated single mutations are overstated. The experiments utilizing tetrad analysis do not meet the standards of yeast genetics and either more tetrads should be analyzed or plasmid shuffle approaches should be used to detect genetic interactions. The authors need to do a better job of explaining why they think that mutant DNA polymerases show defects in both initiation and DNA polymerization. Finally, a more complete discussion of the effects of mutations in genes encoding DNA

polymerases and associated factors on GCR rates is needed. These issues are expanded on below.

Specific Comments

Figure S1A. It would be useful to me if in the crystal structure, there were arrows pointing the the amino acids that define the boundaries of the POPS domain.

P4, L110-123. The authors only test 3 and 5 amino acid substitution mutations. From a cancer perspective, it would be useful to know if any of the single mutations cause a defect. The authors do this later in the paper. They should probably note this here as well as point out that the 3 and 5 amino acid substitution mutations are not really cancer-associated mutations as these are single mutations.

P5, L160-162. The comparison to unknown studies of possibly non all possible mutations is not useful. Just stating the utility of the REL mutant is sufficient.

P6, L231 to P7, L243. I'm not sure I understand the logic that initiation is reduced and DNA polymerization is reduced. In the primer extension assay (Fig 4E) there is less full length and shorter products made. Is this what the authors mean by reduced polymerization. But then why do Figs 4C & D, which show less complete product, support the view that initiation is reduced. Could this just reflect reduced synthesis as seen in the primer extension assay. That there is a defect is clear. But the authors need to do a better job of explaining the logic underlying their conclusion stated on L241-243.

Fig 5A. The authors need to present more tetrads and explain what the cross was and how the genotyping was done. This is a substandard experiment as presented. Alternatively, the authors could construct double mutants containing a URA3 covering plasmid and show the double mutants are sensitive to 5FOA. Also, the authors need to give a reference for the rnh201-RED mutation as its source and established properties are not obvious.

P6, L256. The authors need to indicate that the suppression is partial. Complete suppression is certainly not seen here.

P7, L282-286. It is probably worth referencing the fact that Srivatsan et al PNAS 2019 have reported that mutations affecting PolE increase GCR rates and reported a broad spectrum of many such mutations in different cancers.

P8, L287-288. The authors present no analysis of data supporting this nor do they do the analysis for individual single mutations, which is critical because not all of the single mutations may cause a defect.

P8, L311-331. The single mutant analysis is important but is incomplete and overstated. First, as noted above, this type of tetrad analysis needs to be expanded to be definitive. Nowhere is this clearer than in the case of the L621F mutation where in Fig 6A one smaller colony is shown for the double mutant suggestive of a phenotype. But without more examples, this could just represent colony to colony variation rather than a synthetic growth defect. Consistent with this view, the data in Fig S6A show a very weak phenotype and no other data are presented supportive of a defect. Without further support, it's hard to accept that the L621F mutation causes a significant phenotype. Certainly the E611K mutation causes no phenotype. Finally, the R567C mutation does cause significant defects in the assays presented, but nowhere do the authors accurately state that the phenotypes caused by this mutation are VERY weak compared to the REL mutation. For example, the GCR rate increase is about 1% of that caused by the REL mutation. This leads to an overstated conclusion on L330, 331.

P9, L336-346. I think the conclusions in regard to cancer are overstated. Most of the mutation

analysis is with 3 and 5 amino acid substitution mutations, which are not found in cancer. Of the 3 true cancer associated single mutations studied, only one caused a significant phenotype, which was much weaker than that caused by the 3 and 5 amino acid substitution mutations. Of the other 2 mutations tested, one caused no phenotype and the other caused at best the weakest possible phenotype.

P11, L427, 428. As noted previously, it is not clear why the authors think the mutations affect both initiation and elongation. This needs clarification.

P11, L460-463. As noted above, the results on the cancer associated single mutations are overstated and require clarification. Prior work by Srivatsan et al PNAS 2019 and other studies in which mutations in genes encoding replication factors were shown to increase GCR rates are probably worthy of discussion here.

P12, L472, 473. This sentence seems to be far more speculative than the results on single cancer-associated mutations justifies.

We thank three reviewers for their time in reviewing this work, their positive comments, and helpful suggestions. We have addressed all the points raised by reviewers by including new data, adding clarification, and modification of the text. Our responses and major changes in the text are marked in blue.

Reviewer #1. This study is an exceptional example of how model organisms can be used to explore disease-associated mutations in conserved proteins required for core cellular processes to advance basic research as well as illuminate the underlying mechanisms of disease. This study deserves publication in Nature Communications.

This manuscript describes an impressively comprehensive approach to define a novel functional region of the core catalytic subunit in the major eukaryotic DNA polymerase, Pol ϵ . The core catalytic subunit, Pol2 in yeast and POLE in humans is responsible for leading strand synthesis and has structural and functional features that clearly distinguish it from other DNA Pols, but how these distinct structural features confer this polymerase unique functions in biology is unclear. Mutations in POLE are associated with and likely function as major drivers of some cancers. Understanding the fundamental role of Pol ϵ is thus of critical importance to both basic and health-related research advances. The basic approach was to model specific cancer-associated mutations found in the human ortholog POLE into the yeast ortholog Pol2 and use the experimental power of the yeast model to define the role for this conserved Pol2 region. While this disease-motivated reverse-engineering approach is becoming more standard, this study did an exceptional job and could serve as a primer for how to do this kind of disease-modeling study well. The interesting and sound rationale, the multitude of experimental approaches, and the extremely high quality of the data make this study stand out. The authors focus on non-EXO POLE mutations (whose mechanism and etiology are well studied) found in some cancers that affect a unique conserved region peripheral to the catalytic core. They refer to this region as POPS. POPS is interesting for two major reasons: 1. It is specific to Pol ϵ but its functions are completely unknown; 2. Mutations in this region are associated with specific types cancers, but because there is no mechanistic understanding of this Pol2 region there is no clue as to how these mutations cause cellular defects that promote these cancers.

The authors use extremely sound genetic reasoning and experimentation to engineer yeast where the a Pol2 gene (the catalytic subunit of interest in POLE in humans and Pol2 in yeast) is replaced with a mutant version that encodes three distinct cancer-associated amino acid substitutions, pol2-REL, and show that pol2-REL yeast are temperature-sensitive. (Later they show that single amino acid substitutions, as they are found in human cancers, recapitulate key phenotypes of pol2-REL). These pol2-REL yeast are then used as a tool in subsequent experiments to elucidate how POPS contributes to normal Pol ϵ functions. pol2-REL produced normal levels of Pol2 yet showed numerous novel replication defects, indicating that POPS is important for normal Pol ϵ function and defined a new mechanistic role(s) for Pol2 in genome stability.

A major strength of the study was that multiple independent experiments were used to reinforce important interpretations. For example, direct high-resolution origin mapping and co-IP experiments, in vitro studies, and genetic interaction analyses, each generating beautiful robust data, supported the model that POPS was critical for promoting CMG formation (active form of the helicase) and efficient replication fork progression. These observations in turn were fully consistent with POPS promoting genome stability by suppressing GCRs rather than mutation rate in yeast, and this observation offered an intriguing and reasonable explanation for the etiology of cancers associated with defects in this domain. The basic and disease-relevant aspects of the research were highly complementary. While the rationale and experimental execution of the study were outstanding, there were some issues of the manuscript that could be strengthened. We greatly appreciate reviewer's positive feedbacks and her/his excellent suggestions that can strengthen the manuscript.

In some sections writing was dense, with more emphasis placed on what seemed like relatively minor points than ideal. These issues detracted from the high impact results. For example, there is no need to posit why one of the single amino acid substitutions did not recapitulate the pol2-REL phenotype. What is remarkable is that two did and that the defects make total sense with the new function in Pol2 posited in the model.

This advice is very helpful. We edited throughout the text to make the writing less dense, such as by breaking up long sentences and sections, writing the results more concisely, and better emphasizing high impact results (e.g. page 9 lines 346-347). We also modified abstract to emphasizing the major conclusions.

The title could better emphasize the high impact nature of the work. The title should highlight how the model

organism illuminated the pathology of disease-linked mutations to illuminate both basic and disease-relevant mechanisms.

We agree with reviewer's view on the multi-faceted nature of our work. Our title highlights the mechanistic insights gained from this work, which we believe is the most impactful. We found it difficult to incorporate other aspects of the work into the title within word limits required by the journal. With this said, we have modified discussion to highlight how model organism can help illuminate disease-relevant mechanisms (page 12 lines 505-512).

When possible, the genetics conclusions should be stated in a way to enhance comprehension. For example, consistently use positive statements, i.e. line 26 heading, e.g. "POPS promotes replication initiation and prevents asymmetric fork progression" really means simply "The POPS domain promotes both replication initiation and fork progression". The asymmetric replication structures are merely the data you use to conclude that there is a defect in RF progression from my understanding.

We have changed several section headings to enhance comprehension (e.g. page 4, line 119, 135).

Emphasize that you observe an accumulation of large Y-structures but not small Y-structures. Small Y's would indicate passive replication from a neighboring origin, but you do not see those accumulating. The simplest (only?) explanation is that the two RFs on the fragment are not progressing with equal efficiency, so you see an accumulation of large forks.

This is a very good point and we have modified the text to emphasize this point regarding the difference between large and small Y-shaped replication intermediates (page 5 lines 146-151).

Is it possible given your data that at origins, that only a single CMG is being established, and so at a given origin in a population, those large Y's are a result of only single directional RF forming at an origin (that is, the origin is no longer an origin of bidirectional replication)? It seems this could also result from reduced CMG formation. So the molecular defect is not necessarily always/only because of direct defects in RF progression caused by *pol2-REL* defective *Polε* but rather due to only a single CMG forming in the first place (which of course is ultimately due to the *pol2-REL* defective *Polε*)? If this interpretation is fair, then there are possible two mechanisms by which *Pol2* promotes RF progression--failure to form enough CMGs at initiation and unstable replisomes that have trouble tolerating RF barriers? (Is that what is being conveyed in the Model Fig 6D?)

This is an excellent point that we have also considered. As the reviewer suggested, if reducing CMG levels can result in a single CMG assembly at an origin and subsequent "asymmetric replication initiation", we should detect Y-structures at origins upon lowering CMG levels. We tested this by acute *Dpb11* depletion, an established way to lower CMG levels. We found that this did NOT result in Y-structures (Figure 4B). Hence Y-structures seen in *pol2-REL* cells are unlikely due to lower CMG levels leading to single CMGs assembled at origins and "asymmetric initiation of replication". This interpretation is also in line with the model that a pair of CMGs activates each other to initiate replication as proposed by Mike O'Donnell and others. We have modified text to better described the rationale to test the idea of asymmetric initiation of replication (page 6 lines 231-235).

While no hyper-activation of checkpoints could be detected in *pol2-REL* cells, isn't it difficult to conclude with certainty that the greater defects observed at Late origins are not due to checkpoint signals, below detection, from the many problems that you show are clearly occurring during replication of earlier regions of the genome in *pol2-REL* cells?

We examined two sensitive checkpoint readouts, namely *Rad53* phosphorylation and *RNR4* induction, and found no detectable checkpoint activation in mid-S phase when late origin defects were observed in *pol2-REL* cells. In addition, we provide biochemical data that *pol2-REL* leads to reduced levels of CMG and pre-LC, which explains origin firing defects. Further, our *in vitro* replication assay without the influence of the checkpoint also showed replication initiation defects when the *pol2-REL* protein was used. This combination of complementary data strongly supports our conclusion that *pol2-REL* directly impairs late origin firing. With this said, we added in the text that one cannot exclude the possibility that levels of checkpoint activation below the detection limit of our assays might also contribute to the late origin firing defects in *pol2-REL* cells (page 10 lines 385-387).

The conclusions about the pre-LC defects versus the pre-LC independent defects (i.e. that Dpb11-Pol2 (via POPS) was somewhat confusing and hard to follow. Are you saying that the primary/relevant defect of pol2-REL is not that Pol2 is having trouble joining the pre-LC per se but that its defect interacting with Dpb11 itself is the problem that leads to reduced CMGs and reduced RF progression? It was confusing in how it was highlighted after spending time introducing the pre-LC defects as a possible mechanism.

The reviewer's point is well taken, and we should have provided a clearer explanation. Our data showed that *pol2-REL* is poorly incorporated into the pre-LC, and this can account for the replication initiation defects in *pol2-REL* cells. When Dpb11 is overexpressed in *pol2-REL* cells, replication initiation and CMG levels were improved. Unexpectedly and interestingly, the suppression is not simply caused by better *pol2-REL* incorporation into the pre-LC. Rather, we found increased levels of pair-wise interactions between Dpb11 and *pol2-REL*, between Dpb11 and Sld2 (Fig. 3D), but not between Dpb11 and GINS (Fig. S3C). This finding suggests that there are alternative ways besides pre-LC to enhance CMG assembly when Dpb11 is overexpressed in *pol2-REL* cells. We have included this interpretation in the text (page 10 line 409-410).

Reviewer#2. The manuscript submitted analyses the function of a region of Pol2, the catalytic subunit of DNA Pol ϵ , named by the authors POPS. By introducing some mutations observed cancer, the author suggests that this region plays a role in:

- Origin firing
- Pol ϵ catalytic activity
- Symmetry of fork progression

The authors indicate that the origin firing defects and the defects in fork progression are 2 independent processes that depend on the integrity of POPS. These mutations lead to a sizable increase in gross chromosomal rearrangement, providing an explanation for the chromosomal instability observed in some cancers. This work is interesting and the phenotypes are clearly and convincingly presented. Moreover, these data suggest a novel understanding of the roles of the catalytic domain of Pol2 during origin firing. I think this paper would be of great interest for people in the field. Since the novelty of the model suggested in the paper, however, I believe that more work is required to corroborate the mechanistic model put forward by the authors, thus to support the claims made in the discussion. For this reason I would suggest:

We appreciate reviewer's positive comments of our work. Regarding reviewer's point that we need to corroborate some claims made in the discussion, our intention was to be thought-provoking when discussing the ramifications of our findings. We have now edited the discussion so that our conclusions vs. speculations are more clearly distinguished.

Here we briefly summarize four important conclusions of this work and their implications: i) we uncover a novel and unexpected role for the Pol2 catalytic core in pre-LC assembly; this changes our thinking on how Pol ϵ aids replisome assembly; ii) we demonstrate a critical role of POPS in supporting DNA synthesis; this provides a new perspective on how Pol ϵ acts distinctly from other polymerases to support leading strand synthesis. iii) our data that POPS integrates structural and catalytic roles during the entire S phase; this suggests a dynamic nature of Pol ϵ catalytic core, which is critical to consider when deriving replication models. iv) our data suggests a potential new tumor suppressive effect for human POLE, which is of broad interest.

Major points:

This paper reaches different conclusions from the ones in Muramatsu et al., 2010. I believe this need to be addressed. Araki's group shows, using convincing *in vitro* experiments, that Pol ϵ can bind individually to Dpb11, GINS and Sld2 (especially phospho-Sld2). Also, Dpb2 binds directly GINS (as seen also in Sengupta et al., 2013). Is this the case also in *pol2-REL* (in Fig 2B)? Moreover, GINS binding to Dpb11 and phospho-Sld2 depends on Pol ϵ in Muramatsu et al., 2010. The author suggests instead that Dpb11-Sld2 interact with Psf1 independently of Pol ϵ - although in a non-stoichiometric manner (Fig 3D, S3C) -. The author should address this difference and prove the existence of a Sld2-Dpb11-GINS subcomplex. These data would be more convincing if the author could recapitulate and confirm this model with purified proteins

A follow up paper from the Araki group (Tanaka et al 2013) reported that GINS binds to Dpb11 without Pol ϵ . Reasons for the different conclusions from Tanaka et al vs. Muramatsu et al are unclear, but likely reflect the complexity of the interactions among pre-LC members that we are just beginning to appreciate. Our data is consistent with the conclusion from Tanaka et al that GINS can associate with Dpb11 without Pol ϵ .

Figure 2B addresses the *in vivo* association, not *in vitro* binding, of the tested proteins. It is important to note that *pol2-REL* leads to the surprising revelation that interactions between Dpb11, Sld2 and GINS can

occur without Pol ϵ incorporation. In the discussion we talked about one implication of this finding, which is Dpb11, Sld2 and GINS may form a subcomplex. However, this is not the main conclusion of the current study, and rigorous testing of this model and the multiple ways by which Dpb11, Sld2, GINS, and Pol ϵ can associate with each other and perhaps with additional factors, is a major undertaking for the future. We have modified the text and removed Figure 2C to clarify that a Dpb11-Sld2-GINS subcomplex is one of the interpretations. (page 10 line 421).

Fork asymmetry is among the most striking and interesting aspects of the paper. The authors indicate that this does not depend on the defects in origin firing, since depletion of Dpb11 does not show fork asymmetry. On the other hand, DPB11 overexpression suggests a loss of fork asymmetry in *pol2-REL* (Fig 3B)? Or is this because the experiment is at 37°C? If so, what would happen at 24°C? It would be also interesting to understand where this stalling occurs at ARS 305 and 1212. Could the author use different restriction enzymes so that the replication origin is no longer at the mid-point of a restriction fragment? Are there any specific features that might explain the stalling (e.g. transcription termination)?

The reviewer is correct that in the Dpb11 overexpression experiment, the lack of detectable fork asymmetry is due to experiments being performed at 37°C. We have modified figure legend to be clearer (page 14 lines 561-567). Importantly, our finding that Dpb11 overexpression improves CMG formation and replication initiation levels in *pol2-REL* cells at 37°C provide an explanation for its suppression of *pol2-REL* lethality at 37°C. As the suppression was manifested only at 37°C, repeating the experiment at 24°C is unlikely to further resolve suppression mechanisms.

Specific DNA features that block fork progression lead to high-intensity regions/spots along the Y arc in 2D gels. However, we did not detect such regions at either ARS305 or ARS1212. As suggested by the reviewer, we have now included new data using ARS315, wherein the replication origin is placed asymmetrically in the restriction fragment (new Figure S2C). Again, we observe an increase in large Y structures but no specific high intensity spots. Thus, our data do not indicate fork stalling due to specific chromosome features.

The authors clearly show decreased DNA synthesis in Pol ϵ REL. Is this caused by a decrease in catalytic activity or lower association of Pol ϵ REL to the DNA/replisome and/or an increased turnover?

We demonstrate that overall synthesis rates are reduced in the presence of Pol ϵ^{REL} , relative to Pol ϵ^{WT} , over time and across a range of Pol ϵ concentrations. We also show that Pol ϵ^{REL} has a reduced net catalytic activity. There are many possible reasons for this defect, such as reduced association with DNA or PCNA, diminished dNTP turnover, reduced processivity, or an imbalance between exonuclease and polymerase activities. We believe that a detailed enzymatic characterization of the DNA synthesis defect is beyond the scope of this paper, whereas the net catalytic defect has a significant biological implication, as it can explain the fork progression defects observed in cells using multiple methods. We have expanded discussion on these possibilities (page 11 lines 458-465).

The link between the levels of origin firing and checkpoint activation dynamics has been long established, as seen for example in mutants of SLD2 (Wang H, Elledge SJ 1999) or ORC (Shimada et al., 2002). Moreover, the function of Pol2 in checkpoint activation is known (Navas TA, et al, 1996). The authors claim that *pol2-REL* has defects in origin firing but not checkpoint defects. I would suggest to check whether the dynamics of Rad53 activation are affected in a time course, in strains +/- RAD9.

We have already included the data to show that *pol2-REL* behaves like wild-type for checkpoint activation upon MMS treatment, based on both Rad53 phosphorylation levels and FACS profiles, and that this is drastically different from the control checkpoint mutant allele, *pol2-11*. We have now included time course experiments that demonstrate this point (new Figure S5E). We believe that this data is sufficient to conclude that *pol2-REL* is overall proficient for checkpoint activation. It is conceivable that *pol2-REL* influences on origin firing in normal growth condition may not be the major contributing factor for Pol2 checkpoint function in MMS conditions, which relies on Pol2 C-terminus. We have added this interpretation (page 12, lines 480-483). We note that *pol2-REL* show synthetic lethality with *rad9 Δ* preventing testing in this mutant background (Figure 1 for reviewers); this genetic interaction is consistent with our finding that *pol2-REL* cells turn on G2-M checkpoints as described.

Minor points:

The biochemical experiments are conducted using different conditions (permissive and non-permissive temperatures). This makes somewhat confusing the analysis of the results and the comparison between different experiments. The authors see severe pre-IC defects at 24°C, but a more limited impact on replisome formation (S2E). DPB11 overexpression partially suppress origin firing at 37°C (Fig 3C, S3B) but not pre-IC (120' 37°C). What happens at 24°C (so to easily compare to 2A, B) and in a wild type O/E DPB11? Similar conditions might help comparing results.

As Dpb11 overexpression only suppresses the growth defect of *pol2-REL* cells at 37°C, but not at 24°C, it is more meaningful to determine the molecular effects of Dpb11 overexpression on origin firing and pre-LC levels at 37°C, which we have already included.

In Fig 1D, the early origins, even in wild type, appear to be not really efficient (about 20% fully duplicated). Is my interpretation of the data correct? Why does this look quantitatively different to the data in S1D?

In Figure 1D, the early origin plots are from 30 min after G1 release, a time point when early origin firing defects are most obvious. The profile in Figure S1D was at 40min after G1 release to highlight the late origin firing defects.

Is binding of Pol ε to the CMG affected in *pol2-REL*? Is this the reason why the mutation is synthetic lethal with *dpb2-11*?

The fact that *pol2-REL* or POPS single mutations cause severe growth retardation in *dpb2-1* or *pol32Δ* backgrounds supports the importance of POPS. While the elucidation of the reasons for the synthetic lethality/sickness is of great interest, in our assessment such studies do not affect our main conclusions and should be pursued in the future.

Fig 2D: not sure quantifications reflect what is shown; in wild type the largest amount of Cdc45/Psf1 co-IPing with Mcm4 is at 30' in the film but not in the quantification. The amount of Mcm4, however, appears constant. In addressing Reviewer #1's comments, we have moved this panel to Figure 2A. The quantifications are based on the ratio of Cdc45 or Psf1 levels relative to those of Mcm2 (there is a typo in the label, which is now corrected). For WT cells, the Mcm2 signal at 30 min is much stronger than at 40 min, thus even though Cdc45 or Psf1 signals appeared strongest at 30 min, their ratio to the Mcm2 signal is not the highest. We have included another result using different biological isolate that demonstrate the same point (Figure 2 for reviewers).

I struggle to understand how R-loops (in front of the CMG) might be specifically toxic in a mutant of POL2 that is catalytic deficient. Alternatively, any barrier to fork progression might be a challenge in *pol2-REL* mutants.

What happens, for example, in a *rrm3Δ* background?

We do not mean to say that R-loop is specifically toxic and agree with reviewer that *pol2-REL* likely has difficulty coping with other forms of template barriers as well. We chose to focus on R-loops because their levels can be genetically altered thus allowing testing of how *pol2-REL* responds to R-loop level changes. We were careful to state this and have modified the text to increase clarity (page 7 lines 279-280; page 8 lines 304-307). Supporting this conclusion, we found that *pol2-REL* is synthetically lethal with *rrm3Δ* (new Figure S5A)

Really minor

Line 78-79: sentence not clear (is it the P domain?)

Yes, it is the P domain and we have now pointed this out in the Introduction (page 3 line 74).

line 172: reference repeated

We will remove the repeated reference, thank you.

How did the authors decided to use the mutation S595P? this does not follow the rational of the rest of the mutations used

In the examined POLE domain, five residues show recurrent cancer mutations. While four are highly conserved, S595 of human POLE corresponds to A581 of yeast Pol2. Though Serine and Alanine differ in their R groups, their similarity scores are very close, partly due to their similar size, and thus were included in the initial analysis.

Colour of the crystal structure makes hard to see the POPS region.
We will adjust the color to make POPS more visible.

Psf1 shows an unspecific band in some IPs but not others. An untagged control help making clear which one is the background band.

We note that all experiments were done using untagged Psf1. The extra bands in the IP samples (Fig. 2B/ 3D) when probed for Psf1 (24kDa) correspond to the light chain (25kDa), as we used antibody and Protein G beads during IP. This issue was eliminated when we switched to antibody-conjugated beads in other IPs. We have now added an explanation in the Method section.

Reviewer #3. This is an interesting, mostly well-done study in which the authors examine the role of a small structural domain of PolE/Pol2 in the function of DNA PolE. Through the use of a large number of different assays and an imaginative use of cancer-associated mutations, the authors provide strong evidence for roles of this domain in replisome assembly and DNA synthesis as well as the importance of this domain in vivo. Overall, it is a very solid study that is a worthy contribution to the field.

I do have some concerns that need to be addressed. The results on the cancer-associated single mutations are overstated. The experiments utilizing tetrad analysis do not meet the standards of yeast genetics and either more tetrads should be analyzed or plasmid shuffle approaches should be used to detect genetic interactions. The authors need to do a better job of explaining why they think that mutant DNA polymerases show defects in both initiation and DNA polymerization. Finally, a more complete discussion of the effects of mutations in genes encoding DNA polymerases and associated factors on GCR rates is needed. These issues are expanded on below.

We appreciate reviewer's positive comments and addressed reviewer's specific concerns as detailed below.

Specific Comments

Figure S1A. It would be useful to me if in the crystal structure, there were arrows pointing the amino acids that define the boundaries of the POPS domain.

We have now labeled the examined amino acids at the boundaries of POPS in the crystal structure.

P4, L110-123. The authors only test 3 and 5 amino acid substitution mutations. From a cancer perspective, it would be useful to know if any of the single mutations cause a defect. The authors do this later in the paper. They should probably note this here as well as point out that the 3 and 5 amino acid substitution mutations are not really cancer-associated mutations as these are single mutations.

This is a very good point and we have added this point early in the text as suggested (page 4 lines 121-123).

P5, L160-162. The comparison to unknown studies of possibly non all possible mutations is not useful. Just stating the utility of the REL mutant is sufficient.

The reviewer's point is well taken and we have removed this comparison.

P6, L231 to P7, L243. I'm not sure I understand the logic that initiation is reduced and DNA polymerization is reduced. In the primer extension assay (Fig 4E) there is less full length and shorter products made. Is this what the authors mean by reduced polymerization. But then why do Figs 4C & D, which show less complete product, support the view that initiation is reduced. Could this just reflect reduced synthesis as seen in the primer extension assay. That there is a defect is clear. But the authors need to do a better job of explaining the logic underlying their conclusion stated on L241-243.

The reviewer is correct that primer extension data (in the absence of Pol δ) demonstrates that Pol ϵ^{REL} reduced DNA polymerization. The same conclusion can be drawn in the reconstituted replication system, but only when Pol ϵ was added at a higher concentration ($\geq 30\text{nM}$); this is because Pol δ in the reconstituted replisome system provides compensatory activity for DNA synthesis when Pol ϵ is present at lower concentrations as seen previously (Devbhandari et al, Mol Cell, 2017). So both assays led to the same conclusion.

In the reconstituted replication system wherein proficient strand elongation was seen with low levels of Pol ϵ^{REL} , total DNA synthesis under these conditions reflect origin activation. Time course tests done using 15 nM Pol ϵ showed that overall DNA synthesis was reduced by up to ~40 % in the presence of Pol ϵ^{REL} .

compared with Pol ϵ^{WT} , indicating a reduced rate of origin activation. We have modified this section to better described the results.

Fig 5A. The authors need to present more tetrads and explain what the cross was and how the genotyping was done. This is a substandard experiment as presented. Alternatively, the authors could construct double mutants containing a URA3 covering plasmid and show the double mutants are sensitive to 5FOA. Also, the authors need to give a reference for the *rnh201-RED* mutation as its source and established properties are not obvious.

We have now included five tetrads for each cross in the new Figure S5A and explained the crosses and genotyping in the methods (page 18 lines 742-747). We have included the reference for *rnh201-RED* (page 8 lines 287-288).

P6, L256. The authors need to indicate that the suppression is partial. Complete suppression is certainly not seen here.

Done (page 8 line 288).

P7, L282-286. It is probably worth referencing the fact that Srivatsan et al PNAS 2019 have reported that mutations affecting PolE increase GCR rates and reported a broad spectrum of many such mutations in different cancers.

Srivatsan et al 2019 examined the *pol2-12* checkpoint allele in yeast GCR assay, and surveyed POLE mutations in cancers. We note that *pol2-12* affects the C-terminal end of Pol2 involved in checkpoint response, and no corresponding POLE mutations have been reported in cancer patients. In contrast, POPS mutations were derived from cancer patient mutations and affect the uncharacterized Pol2 N-terminal domain within the catalytic core. There is a clear difference between these two studies. With this said, we have included Srivatsan et al 2019 in the discussion (page 12 lines 496-497).

P8, L287-288. The authors present no analysis of data supporting this nor do they do the analysis for individual single mutations, which is critical because not all of the single mutations may cause a defect.

We used the classical *CAN1* loss-of-function assay to measure mutation rates and found no difference between wild-type and *pol2-REL* cells. To our knowledge, this assay has reported increased mutation rates for all known mutators by many labs. We now describe this assay in more detail and knowledge that the assay does not report mutations that do affect the *CAN1* gene function (page 8, lines 318-320).

P8, L311-331. The single mutant analysis is important but is incomplete and overstated. First, as noted above, this type of tetrad analysis needs to be expanded to be definitive. Nowhere is this clearer than in the case of the L621F mutation where in Fig 6A one smaller colony is shown for the double mutant suggestive of a phenotype. But without more examples, this could just represent colony to colony variation rather than a synthetic growth defect. Consistent with this view, the data in Fig S6A show a very weak phenotype and no other data are presented supportive of a defect. Without further support, it's hard to accept that the L621F mutation causes a significant phenotype. Certainly the E611K mutation causes no phenotype. Finally, the R567C mutation does cause significant defects in the assays presented, but nowhere do the authors accurately state that the phenotypes caused by this mutation are VERY weak compared to the REL mutation. For example, the GCR rate increase is about 1% of that caused by the REL mutation. This leads to an overstated conclusion on L330, 331.

We have now included new Figure S6A to show 6 tetrads per diploid and explained genotyping in the methods (page 18 lines 742-747). Our conclusion for *pol2-R567C* and *pol2-4* remained the same. In addition, *pol2-L621F* clearly showed synthetic sickness with *dpb2-1* as we have already stated, through its interaction with *pol32Δ* is unclear. We have adjusted the text accordingly (page 9 lines 350-351). FACS profile difference between *pol2-L621F* and WT in Figure S6A (now Figure S6B) is highly reproducible; we included the data using a different biological isolate to show the same point (Figure 3 for reviewers).

We have subjected *pol2-R567C* to several assays and in each one it recapitulated *pol2-REL* phenotype. These include i) slow S phase progression (FACS), ii) replication initiation delay (2D gel), iii) replication fork asymmetric movement (2D gel), iv) negative genetic interactions with *pol32Δ* and *dpb2-1*, and v) increased GCR rates. It is expected that *pol2-R567C* is less defective than *pol2-REL*, and we have adjusted the sentence

to reflect this better (page 9 line 359). However, we argue that it is more important to demonstrate that *pol2-R567C* differs from wild-type in multiple assays.

P9, L336-346. I think the conclusions in regard to cancer are overstated. Most of the mutation analysis is with 3 and 5 amino acid substitution mutations, which are not found in cancer. Of the 3 true cancer associated single mutations studied, only one caused a significant phenotype, which was much weaker than that caused by the 3 and 5 amino acid substitution mutations. Of the other 2 mutations tested, one caused no phenotype and the other caused at best the weakest possible phenotype.

We were careful in talking about the implication of the work to cancer in the discussion section. POPS contains multiple recurrent cancer mutations besides the three examined here. We meant to highlight the general consequence of POPS perturbation and its relevance to genome instability captured in cancer genomes.

P11, L427, 428. As noted previously, it is not clear why the authors think the mutations affect both initiation and elongation. This needs clarification.

We have addressed this point above.

P11, L460-463. As noted above, the results on the cancer associated single mutations are overstated and require clarification. Prior work by Srivatsan et al PNAS 2019 and other studies in which mutations in genes encoding replication factors were shown to increase GCR rates are probably worthy of discussion here.

We have addressed this point above.

P12, L472, 473. This sentence seems to be far more speculative than the results on single cancer-associated mutations justifies.

We indicate that this is a speculation.

Figure 1 for reviewers

Figure 2 for reviewers

Figure 3 for reviewers

REVIEWERS' COMMENTS:

Reviewer #1 (Remarks to the Author):

The authors have addressed each of my concerns and I am satisfied. This important study should be published

Reviewer #2 (Remarks to the Author):

The revised text is OK. This paper raises many open questions but I agree with the author that addressing these might go beyond the scope of this paper. The reply to my review is reasonable. I therefore support the publication of this paper.

(PS: As the author might have noticed, in several figure the error bars appear not aligned/shifted; this will needs fixing before publication.)

Reviewer #3 (Remarks to the Author):

The authors have adequately addressed all of my comments through revisions to the text and the addition of new data. I have no further significant concerns.

There are some typos and awkward sentences so I suggest the authors proof read their manuscript one last time. Examples include:

P7, L257, 258 where "...can be due to that..." should be replaced by "...could occur if...".

P9, L359 where "rearrangement" should be "re-arrangements".

Response to REVIEWERS' COMMENTS:

Reviewer #1. The authors have addressed each of my concerns and I am satisfied. This important study should be published

Reviewer #2. The revised text is OK. This paper raises many open questions but I agree with the author that addressing these might go beyond the scope of this paper. The reply to my review is reasonable. I therefore support the publication of this paper.

(PS: As the author might have noticed, in several figure the error bars appear not aligned/shifted; this will need fixing before publication.)

We have adjusted the alignment.

Reviewer #3. The authors have adequately addressed all of my comments through revisions to the text and the addition of new data. I have no further significant concerns. There are some typos and awkward sentences so I suggest the authors proof read their manuscript one last time.

Examples include: P7, L257, 258 where "...can be due to that..." should be replaced by "...could occur if...".

We have adjusted this sentence.

P9, L359 where "rearrangement" should be "re-arrangements".

The suggested correction is made.